# FORMULATING GENERALIZABLE AND NON-GENERALIZABLE INTERACTIONS IN DNNS

## ABSTRACT

This paper aims to analyze the generalization power of deep neural networks (DNNs) from the perspective of interactions. Unlike previous analyses of a DNN's generalization power in a high-dimensional feature space, we find that the generalization power of a DNN can be explained as the generalization power of the interactions. We find that generalizable interactions follow a decay-shaped distribution, while non-generalizable interactions follow a spindle-shaped distribution. Furthermore, we develop a method to disentangle these two types of interactions in a DNN. We have verified that the theoretically disentangled distributions of generalizable interactions and non-generalizable interactions can well match the real distributions in experiments.

## 1 INTRODUCTION

Analyzing and quantifying the generalization power of deep neural networks (DNNs) is a critical issue in deep learning. Prior studies have investigated the generalization power of a DNN from multiple perspectives, including analyzing the generalization bound *w.r.t.* the loss gap (Neyshabur et al., 2017; Bousquet et al., 2020), assessing the smoothness of the loss landscape (Keskar et al., 2016; Li et al., 2018), and using neural tangent kernel (NTK) theory (Jacot et al., 2018).

In contrast to previously treating generalization power as an intrinsic property of the entire model, an emerging trend in this field is to provide a more rigorous and nuanced explanation of DNN generalization. A fundamental question has plagued the field for years: *Can the generalization power of **an entire DNN** be explained through the generalizability of its **compositional inference patterns**?*

**Background: symbolic generalization.** In response to this question, symbolic generalization has been proposed (Ren et al., 2024a; Zhou et al., 2024; Chen et al., 2024) as a new explanation strategy and has received significant attention. Symbolic generalization primarily focuses on two key aspects. (1) Despite appearing seemingly counterintuitive, it has been proven (Ren et al., 2024a) that **the highly complex primitive inference patterns in a DNN can be rigorously formulated into a set of symbolic/sparse interaction patterns.** (2) In this way, it enables people to define and use the generalizability of these interactions to explain the performance of an entire DNN.

Specifically, given an input sample, people can extract a set of AND/OR interactions from the DNN. As shown in Figure 1, each interaction represents an AND relationship (or an OR relationship) between input variables, which is used by the DNN for inference. For instance, an AND interaction between tokens in $S = \{theory, relativity\}$ is activated only when both tokens co-occur in the input, and it contributes a numerical effect $I_S^{\text{AND}} = 0.30$ to boost the confidence of generating "space." Crucially, Ren et al. (2024a) have proven that *these interaction effects can precisely predict network outputs across exponentially many samples, which ensures the scientific rigor of the explanation.*

**Our work:** In this paper, we further investigate another fundamental issue in symbolic generalization, *i.e.*, *how to use the generalizability of compositional interaction patterns to explain the generalization power of an entire DNN*. To this end, the generalizability of an interaction to testing samples is defined by Chen et al. (2024) as the capacity of the interaction being used to classify testing samples. Previous studies have investigated the generalization power (Zhou et al., 2024) and the robustness of the interactions (Liu et al., 2023; Ren et al., 2023b).

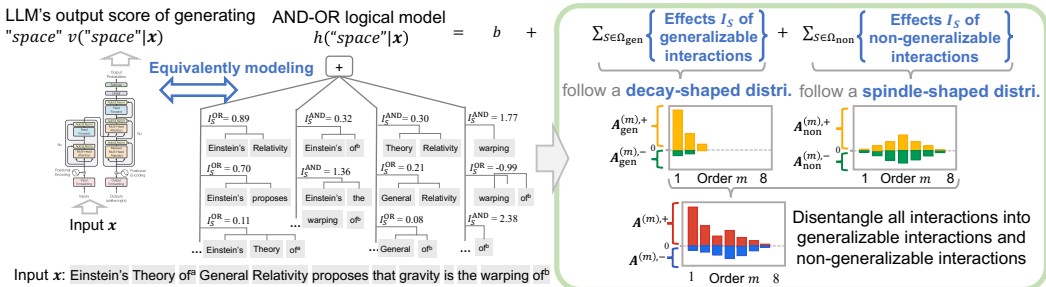

Figure 1: (Left) A series of theorems (Ren et al., 2023a; Li & Zhang, 2023; Ren et al., 2024a) have proven that the complex inference patterns of a DNN on a given input can be represented in mathematics by a logical model based on AND-OR interactions. (Right) Our findings reveal that interactions in a DNN can be categorized into a set of generalizable interactions following a decay-shaped distribution and a set of non-generalizable interactions following a spindle-shaped distribution. **Please see Figure 6 in Appendix B for examples of AND-OR logical model that explains LLMs.**

However, despite abundant empirical findings, progress in the field of symbolic generalization is hindered by a more fundamental issue, *i.e.*, **an analytical formulation of the generalizable interactions and non-generalizable interactions**.

Specifically, in this paper, we discover and further prove that generalizable interactions and non-generalizable interactions follow two distinct distributions over different complexities. The complexity of an interaction $S$ is measured by its order, *i.e.*, the number of input variables in $S$, $order(S) = |S|$.

(1) **Generalizable interactions encoded by a DNN exhibit a characteristic decay-shaped distribution with increasing orders (complexities)** (see Figure 1 (right)). This indicates that most generalizable interactions appear in lower orders, and the interaction strength diminishes as the order increases. Notably, the discovered decay-shaped distribution provides a more precise characterization of generalizable interactions, moving beyond the coarse notion of simplicity bias, *i.e.*, the "best" representation does not merely refer to the "simplest" ones.

(2) **Non-generalizable interactions typically exhibit a spindle-shaped distribution across different orders (complexities)**. Figure 1 (right) shows that most non-generalizable interactions are of medium orders, with a small proportion of non-generalizable interactions appearing in low and extremely high orders. Additionally, non-generalizable interactions exhibit mutual cancellation effects between their positive and negative components.

**Disentangling the generalizable and non-generalizable interactions.** Therefore, we further propose a method to formulate and disentangle generalizable interactions in a decay-shaped distribution and non-generalizable interactions in a spindle-shaped distribution. Experimental results show that our method accurately captures the true distributions of generalizable and non-generalizable interactions.

**Theoretical and practical value.** The derived distributions of generalizable and non-generalizable interactions in DNNs have demonstrated significant application potential and theoretical value. First, our theory enables people to directly analyze a DNN's representation quality even without a need for numerous testing samples. This provides a new mathematical tool to identify representational deficiencies in DNNs, even from correctly classified samples (please see Figure 8 for results). Thus, our method significantly reduces the computational cost for sample collection when we evaluate a DNN's generalization power.

Second, our theory has clarified several ambiguous conclusions in mathematics. For instance, our theory rectifies the vague assertion presented in (Zhou et al., 2024) regarding the poor generalization power of middle-order and high-order interactions, by explicitly characterizing the distribution of such non-generalizable interactions.

The contributions of this paper can be summarized as follows. We establish that generalizable interactions encoded by a DNN typically follow a decay-shaped distribution, whereas non-generalizable interactions tend to follow a spindle-shaped distribution. Furthermore, we propose a method to formulate and disentangle interactions of a spindle-shaped distribution and interactions of a decay-shaped distribution from a DNN. Extensive experiments have validated the effectiveness of our method.

## 2 ANALYZING INTERACTIONS OF TWO DISTRIBUTIONS

### 2.1 PRELIMINARIES: INTERACTIONS REPRESENT PRIMITIVE INFERENCE PATTERNS IN DNNs

Let us consider a DNN and an input sample $\boldsymbol{x} = [x_1, x_2, \ldots, x_n]$ composed of $n$ input variables[1]. Let $N = \{1, 2, \ldots, n\}$ denote the indices of the $n$ input variables. The scalar output of the DNN, $v(\boldsymbol{x}) \in \mathbb{R}$, is typically defined as the classification confidence[2] of the sample $\boldsymbol{x}$, following Deng et al. (2022):

$$v(\boldsymbol{x}) \stackrel{\text{def}}{=} \log \frac{p(y = y^{\text{truth}}|\boldsymbol{x})}{1 - p(y = y^{\text{truth}}|\boldsymbol{x})} \in \mathbb{R}, \tag{1}$$

where $p(y = y^{\text{truth}}|\boldsymbol{x})$ represents the probability[2] of classifying $\boldsymbol{x}$ into the ground-truth category $y^{\text{truth}}$.

**Problem setting:** To analyze the generalization power of a DNN from the perspective of detailed inference patterns, studies of symbolic generalization require extracting inference patterns encoded by the DNN. To this end, a logical model $h$ is introduced to explain the inference patterns of the DNN $v$. This model must satisfy two critical yet seemingly conflicting requirements to ensure the faithfulness of the explanation. **(1) Fidelity requirement:** the model $h$ must accurately predict the DNN's outputs over a large input samples set $\mathcal{D}$. **(2) Conciseness requirement:** the model $h$ should be based on sufficiently simple logic to ensure its interpretability. These requirements are formally expressed as follows:

$$\forall x' \in \mathcal{D}, \quad h(\boldsymbol{x}') = v(\boldsymbol{x}'), \quad \text{subject to} \quad \text{complexity}(h) \leq M, \tag{2}$$

where $M$ is an upper bound on the complexity of the logical model $h$.

Specifically, the model $h$ is formulated as an AND-OR logical model, which encodes a set of AND-OR interaction logic as illustrated in Figure 1. The universal matching property and the sparsity property are later proven to satisfy the fidelity requirement and the conciseness requirement, respectively. **Please refer to the video demo in the supplementary material for the symbolic generalization. Figure 6 shows logical models to explain LLMs.**

$$h(\boldsymbol{x}_{\text{mask}}) \stackrel{\text{def}}{=} \sum_{S \in \Omega^{\text{AND}}} \mathbb{1}_{\text{AND}}(S|\boldsymbol{x}_{\text{mask}}) \cdot I_S^{\text{AND}} + \sum_{S \in \Omega^{\text{OR}}} \mathbb{1}_{\text{OR}}(S|\boldsymbol{x}_{\text{mask}}) \cdot I_S^{\text{OR}} + b \tag{3}$$

*The trigger function* $\mathbb{1}_{\text{AND}}(S|\boldsymbol{x}_{\text{mask}})$ represents an AND interaction between a subset $S \subseteq N$ of input variables. If all input variables in $S$ are present (not masked) in the masked sample $\boldsymbol{x}_{\text{mask}}$[3], the AND trigger function is activated and returns 1. Otherwise, it returns 0. *The trigger function* $\mathbb{1}_{\text{OR}}(S|\boldsymbol{x}_{\text{mask}})$ represents an OR interaction between a subset $S \subseteq N$ of input variables. If at least one input variable in $S$ is present (not masked) in $\boldsymbol{x}_{\text{mask}}$, then the OR trigger function is activated and returns 1. Otherwise, it returns 0. $I_S^{\text{AND}}$ and $I_S^{\text{OR}}$ are the scalar weights. The bias term $b$ is set to $v(\boldsymbol{x}_\emptyset)$, *i.e.*, the network output when masking all input variables in $\boldsymbol{x}$. $\Omega^{\text{AND}}$ and $\Omega^{\text{OR}}$ represent the set of AND interactions and the set of OR interactions, respectively.

**First, the fidelity requirement is satisfied by the following universal matching property in Theorem 2.1.** It shows that, with a specific setting of weights $I_S^{AND}$ and $I_S^{OR}$, the logical model $h(\cdot)$ can always predict the DNN's outputs $v(\cdot)$ on an exponential number of masked samples. The sample set $\mathcal{D} = \{\boldsymbol{x}_T | T \subseteq N\}$ is thus implemented as $2^n$ masked states of the input sample $\boldsymbol{x}$.

**Theorem 2.1.** *(Universal matching property, proved in (Chen et al., 2024) and Appendix D) Given an input sample $\boldsymbol{x}$, if all weights $\forall S \subseteq N$ are set as $I_S^{AND} = \sum_{T \subseteq S}(-1)^{|S|-|T|} \cdot u_T^{AND}$ and $I_S^{OR} = -\sum_{T \subseteq S}(-1)^{|S|-|T|} \cdot u_{N \setminus T}^{OR}$, then the following holds:*

$$\forall T \subseteq N, \quad v(\boldsymbol{x}_T) = h(\boldsymbol{x}_T), \tag{4}$$

*where $\boldsymbol{x}_T$ denotes the masked sample retaining only the input variables in $T$, while input variables in the set $N \setminus T$ are masked.[3] The overall network output $v(\boldsymbol{x}_T)$ is decomposed according to $u_T^{AND} = 0.5 \cdot v(\boldsymbol{x}_T) + \gamma_T$ and $u_T^{OR} = 0.5 \cdot v(\boldsymbol{x}_T) - \gamma_T$. $\{\gamma_T\}$ is a set of learnable parameters to determine the decomposition (see Appendix F for the details).*

---

[1]See Appendix K.2 for details on treating token embeddings or image patches as individual input variables.

[2]See Appendix K.2 for the probability in tasks of language generation and image classification.

[3]Input variables $i$ are typically masked by setting $\boldsymbol{x}_i$ to baseline values $b_i$. In image classification, we follow Dabkowski & Gal (2017) to set $b_i$ as the average color value across all pixels in all images. For LLMs, we follow Shen et al. (2023) to set $b_i$ as a special token (*i.e.,* [MASK] token).

*How to extract interactions.* In order to obtain interaction effects $I_S^{\text{AND}}$ and $I_S^{\text{OR}}$ for all subsets $S \subseteq N$, Li & Zhang (2023) proposed to optimize parameters $\{\gamma_T\}$ defined in Theorem 2.1 by minimizing the function $\sum_S |I_S^{\text{AND}}| + \sum_S |I_S^{\text{OR}}|$. Please see Appendix F for the interaction extraction pseudo-code.

**Second, the conciseness requirement is satisfied by the sparsity[4] property of interactions.** As proved by Ren et al. (2024a), a well-trained DNN[5] encodes only a few interactions. Specifically, only $O(n^p/\tau)$ interactions have salient effects *s.t.* $|I_S^{\text{AND/OR}}| > \tau$, where $p \in [1.5, 2]$ in most cases. All other interactions have negligible effects $I_S^{\text{AND/OR}} \approx 0$. Here, $\tau$ is a threshold for salient interactions.

*Therefore, the above requirements ensure that we can construct a sufficiently concise logical model $h$ comprising only a few salient interactions by setting $\Omega^{AND} = \{S \subseteq N : |I_S^{AND}| > \tau\}$ and $\Omega^{OR} = \{S \subseteq N : |I_S^{OR}| > \tau\}$. This model can accurately predict the DNN's outputs across all $2^n$ masked states of an input sample $x \in \mathcal{D}$, thereby providing an objective explanation of the DNN. This has been empirically verified on diverse DNNs in (Li & Zhang, 2023; Ren et al., 2023b; Zhou et al., 2024).*

**Order/complexity of interactions.** The order (or complexity) of an interaction $S$ is defined as the number of input variables in the set $S$, *i.e., order$(S) = |S|$.*

## 2.2 Quantifying generalizable and non-generalizable interactions

Inspired by the fact that a DNN's output can be faithfully decomposed into the effects of AND-OR interactions (see Theorem 2.1 and examples in Figure 6), explaining a DNN's performance through the generalizability of interactions has emerged as a new research direction in recent years, namely *symbolic generalization*. Numerous empirical studies have contributed to this direction:

- Zhou et al. (2024) showed that middle-order and high-order interactions typically generalized less effectively than low-order interactions.

- Ren et al. (2024b) found that DNNs mainly learned high-order interactions in the overfitting phase.

- Liu et al. (2023); Ren et al. (2023b) observed that high-order interactions were less robust to feature noise compared to low-order interactions.

However, in this study, **we focus on a more fundamental question, *i.e.*, how to mathematically formulate generalizable interactions and non-generalizable interactions**, rather than merely relying on isolated empirical observations.

**Definition of generalization power of interactions.** Zhou et al. (2024) have defined the generalizable interactions as follows. For each salient AND/OR interaction, if it frequently appears on testing samples, and exhibits similar effects on the classification confidence, then this interaction is considered to generalize to testing samples; otherwise, not. For example, the interaction between $S = \{\text{nose}, \text{L-eye}, \text{R-eye}\}$ can usually generalize to different face images for face detection.

**Metric for generalization power of interactions.** However, the above definition requires prohibitive computational cost of exhaustively searching interactions through testing samples. Thus, we use and further extend an approximate yet more efficient method (He et al., 2025). This method trains an additional DNN on testing samples, referred to as the **baseline DNN**. This method then considers all interactions shared by both target DNN $v$ and the baseline DNN $v^{\text{test}}$ generalizable to testing samples. It is because all interactions (denoted by $\{I_S^{\text{AND,test}}, I_S^{\text{OR,test}}\}$) in $v^{\text{test}}$ are learned from testing samples. In this way, given a salient AND interaction *s.t.* $|I_S^{\text{AND}}| > \tau$, its generalizability can be identified by the following binary metric $\rho_S^{\text{AND}} \in \{0, 1\}$, which requires that this interaction be also salient in $v^{\text{test}}$ (*i.e.*, $|I_S^{\text{AND,test}}| > \tau$) and make similar effect $I_S^{\text{AND}} \cdot I_S^{\text{AND,test}} > 0$.

$$\rho_S^{\text{AND}} = \mathbb{1}(|I_S^{\text{AND,test}}| > \tau \text{ and } I_S^{\text{AND}} \cdot I_S^{\text{AND,test}} > 0), \quad \rho_S^{\text{OR}} = \mathbb{1}(|I_S^{\text{OR,test}}| > \tau \text{ and } I_S^{\text{OR}} \cdot I_S^{\text{OR,test}} > 0). \quad (5)$$

The generalizability of an OR interaction $\rho_S^{\text{OR}}$ is defined in the same manner.

Then, we propose to use a set of $k$ baseline DNNs, denoted by $\{v_i\}_{i=1}^k$, to extend the metrics of $\rho_S^{\text{AND}}$ and $\rho_S^{\text{OR}}$ and obtain more convincing measurement of generalization. Because we cannot ensure a

---

[4]In the field of symbolic generalization, the sparsity is defined as the state that almost all interactions have negligible values with only very few interactions having salient values. Please see Appendix C for details.

[5]The sparsity of interactions can be guaranteed by three common conditions for the DNN's smooth inferences on randomly masked samples. Detailed conditions are provided in the Appendix C.

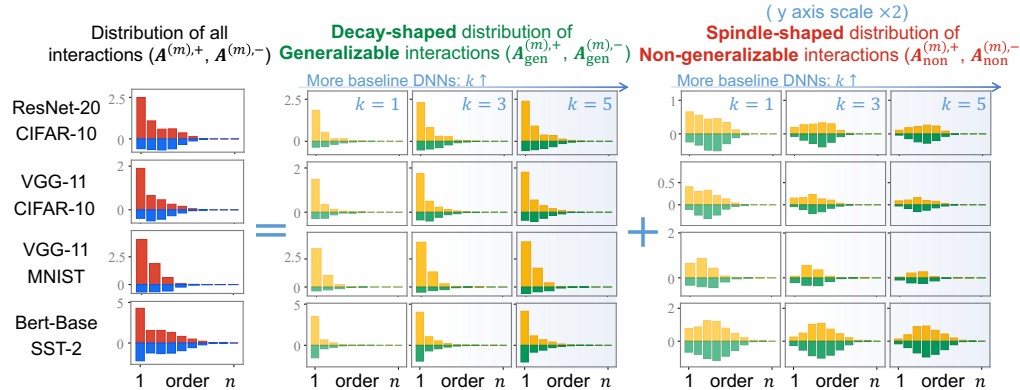

Figure 2: Generalizable interactions ($\mathbf{A}_{\text{gen}}^{(m),+}$ and $\mathbf{A}_{\text{gen}}^{(m),-}$) exhibit a decay-shaped distribution, while non-generalizable interactions ($\mathbf{A}_{\text{non}}^{(m),+}$ and $\mathbf{A}_{\text{non}}^{(m),-}$) exhibit a spindle-shaped distribution. Furthermore, ablation studies show that when using three baseline DNNs (*i.e.*, $k = 3$), the two distributions have already converged (close to those observed when $k = 5$). To enhance clarity, we double the y-axis scale for non-generalizable interaction distributions. **More results are provided in Appendix L.1.**

single baseline DNN to encode all potential interactions, the generalizable interaction is redefined by the generalizability to at least one baseline DNN, $\hat{\rho}_S^{\text{AND}} = \mathbb{1}(\rho_{S,v_1}^{\text{AND}} = 1) \vee \cdots \vee \mathbb{1}(\rho_{S,v_k}^{\text{AND}} = 1)$ and $\hat{\rho}_S^{\text{OR}} = \mathbb{1}(\rho_{S,v_1}^{\text{OR}} = 1) \vee \cdots \vee \mathbb{1}(\rho_{S,v_k}^{\text{OR}} = 1)$, where $\rho_{S,v_i}^{\text{AND}}$ and $\rho_{S,v_i}^{\text{OR}}$ are defined in Equation (5) and represents whether the AND/OR interaction $S$ transfers to the baseline DNN $v_i$.

**Quantifying the distributions of generalizable and non-generalizable interactions.** Given a DNN, we use the following four metrics to measure the total strength of positive interactions of the $m$-th order $\mathbf{A}^{(m),+}$, the total strength of negative interactions of the $m$-th order $\mathbf{A}^{(m),-}$, the strength of generalizable positive interactions of the $m$-th order $\mathbf{A}_{\text{gen}}^{(m),+}$ and the strength of generalizable negative interactions of the $m$-th order $\mathbf{A}_{\text{gen}}^{(m),-}$:

$$\mathbf{A}^{(m),+} = \sum_{\substack{S \in \Omega^{\text{AND}} \\ |S|=m}} \max\{I_S^{\text{AND}}, 0\} + \sum_{\substack{S \in \Omega^{\text{OR}} \\ |S|=m}} \max\{I_S^{\text{OR}}, 0\}, \quad \mathbf{A}_{\text{gen}}^{(m),+} = \sum_{\substack{S \in \Omega^{\text{AND}} \\ |S|=m}} \hat{\rho}_S^{\text{AND}} \max\{I_S^{\text{AND}}, 0\} + \sum_{\substack{S \in \Omega^{\text{OR}} \\ |S|=m}} \hat{\rho}_S^{\text{OR}} \max\{I_S^{\text{OR}}, 0\},$$

$$\mathbf{A}^{(m),-} = \sum_{\substack{S \in \Omega^{\text{AND}} \\ |S|=m}} \min\{I_S^{\text{AND}}, 0\} + \sum_{\substack{S \in \Omega^{\text{OR}} \\ |S|=m}} \min\{I_S^{\text{OR}}, 0\}, \quad \mathbf{A}_{\text{gen}}^{(m),-} = \sum_{\substack{S \in \Omega^{\text{AND}} \\ |S|=m}} \hat{\rho}_S^{\text{AND}} \min\{I_S^{\text{AND}}, 0\} + \sum_{\substack{S \in \Omega^{\text{OR}} \\ |S|=m}} \hat{\rho}_S^{\text{OR}} \min\{I_S^{\text{OR}}, 0\}.$$

(6)

Based on the above metrics, we further quantify the distributions of non-generalizable interactions by $\mathbf{A}_{\text{non}}^{(m),+} = \mathbf{A}^{(m),+} - \mathbf{A}_{\text{gen}}^{(m),+}$ and $\mathbf{A}_{\text{non}}^{(m),-} = \mathbf{A}^{(m),-} - \mathbf{A}_{\text{gen}}^{(m),-}$. To this end, for a specific order $m$, if the values of $\mathbf{A}_{\text{gen}}^{(m),+}$ and $\mathbf{A}_{\text{gen}}^{(m),-}$ are high relative to $\mathbf{A}^{(m),+}$ and $\mathbf{A}^{(m),-}$, respectively, then the $m$-order interactions exhibit high generalization power; otherwise, not.

## 2.3 DISTRIBUTIONS OF GENERALIZABLE AND NON-GENERALIZABLE INTERACTIONS

In this subsection, we visualize the distribution of generalizable interactions and that of non-generalizable interactions in Figure 2. The visualized distributions inspire us to propose the following two empirical claims, and we analyze the two claims from various perspectives. More crucially, we will further formulate the interaction distributions in Section 3.

**Claim 1. Generalizable interactions** follow a decay-shaped distribution: they are predominantly concentrated in lower orders, and their strength decreases monotonically with increasing orders.

**Claim 2. Non-generalizable interactions** follow a spindle-shaped distribution: they are primarily concentrated in middle orders, with minimal presence in both low orders and extremely high orders. Additionally, they exhibit mutual cancellation effects between their positive and negative components.

*Implementation details of visualizing the distribution.* We trained VGG-11 (Simonyan & Zisserman, 2014) and ResNet-20 (He et al., 2016) on the MNIST dataset (LeCun et al., 1998) and CIFAR-10 dataset (Krizhevsky et al., 2009). We also trained BERT-Medium and BERT-Base models Devlin et al. (2019) on the SST-2 dataset Socher et al. (2013). We followed the experimental settings in

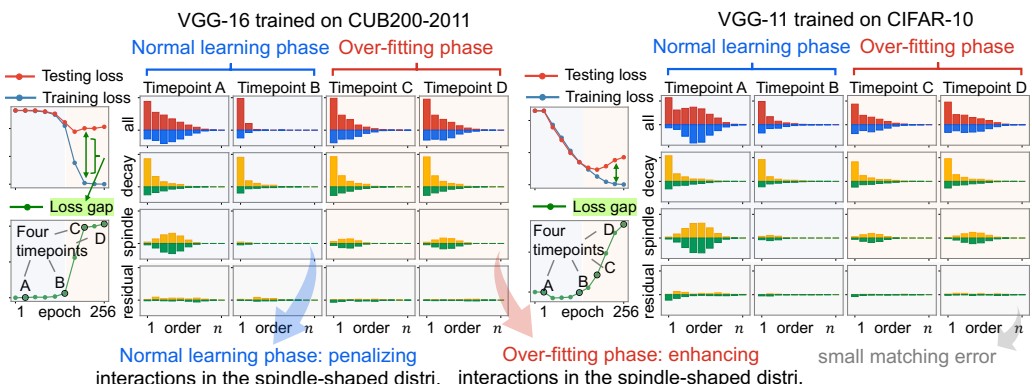

Figure 3: The two-stage dynamics of interactions in the learning process of a DNN. The first row illustrates the change of interactions ($\mathbf{A}^{(m),+}$ and $\mathbf{A}^{(m),-}$) throughout the entire learning process. The remaining three rows visualize the distributions of the disentangled generalizable interactions ($\mathcal{A}_{\text{decay}}^{(m),+}$ and $\mathcal{A}_{\text{decay}}^{(m),-}$), non-generalizable interactions ($\mathcal{A}_{\text{spindle}}^{(m),+}$ and $\mathcal{A}_{\text{spindle}}^{(m),-}$), and the residual term ($\mathcal{A}_{\text{residual}}^{(m),+}$ and $\mathcal{A}_{\text{residual}}^{(m),-}$). **More results across different DNNs and datasets are provided in Appendix L.2.**

(Ren et al., 2024b) to sample image patches as input variables for image models, and used the entire embedding vector of a token as a single input variable for language models. We set $k = 3$ in all subsequent experiments, because we found the setting $k = 3$ empirically yielded more reliable results than other settings $k = 1$ and $k = 5$. More details can be found in Appendix K.2.

**Developing and revising the previous claim on the generalization power of interactions.** The above claims specify the explicit distributions of generalizable interactions and non-generalizable interactions. This further refines previous discovery on the generalization power of interactions in (Zhou et al., 2024), *i.e.*, the generalization power of middle-order and high-order interactions is generally lower than that of low-order interactions.

We find that the more accurate claim is that non-generalizable interactions predominantly exhibit a spindle-shaped distribution, whereas generalizable interactions typically follow a decay-shaped pattern, primarily comprising low-order interactions.

**Echoing the finding of the two-stage dynamics of interactions.** Our findings of the change of interactions' generalizability in Figure 3 are consistent with and explain the two-stage dynamics of interactions reported by Zhang et al. (2024); Ren et al. (2024b). Specifically, the entire training process of a DNN is found to contain two phases: (1) *During the normal learning phase*, the testing loss remains close to the training loss, and the DNN primarily eliminates high-order interactions and learns low-order interactions. (2) Subsequently, *in the overfitting phase*, the training-testing loss gap suddenly increases, and the DNN begins to learn interactions of progressively higher orders.

Thus, we track and visualize the change of interaction distributions ($\mathbf{A}^{(m),+}$ and $\mathbf{A}^{(m),-}$) during the learning process in the first row of Figure 3. (1) During the normal learning phase, the DNN eliminates non-generalizable interactions in a spindle-shaped distribution (Timepoint A and B), retaining only those with a decay-shaped distribution. (2) In the overfitting phase (Timepoints C and D), the DNN re-learns the non-generalizable interactions in the spindle-shaped distribution.

## 3 DISENTANGLING TWO DISTRIBUTIONS OF INTERACTIONS

### 3.1 MODELING THE SPINDLE-SHAPED DISTRIBUTION

Experimental results in Section 2.3 reveal that non-generalizable interactions typically exhibit a spindle-shaped distribution characterized by the cancellation between positive and negative interaction effects. However, the underlying rationale and mathematical property of this distribution remain unexplored. The only relevant finding is provided by Zhang et al. (2024), who proved that each interaction extracted from a model with fully random (noise) outputs follows a Gaussian distribution.

Therefore, we aim to formulate the distribution of non-generalizable interactions. To this end, we extend the findings of Zhang et al. (2024) to the assumption that non-generalizable feature components in a DNN would add Gaussian noise to each interaction. Based on this, we prove that when measuring the strength of all $m$-order non-generalizable interactions, the distributions of $\mathbf{A}^{(m),+}$ and $\mathbf{A}^{(m),-}$ over different orders $m$ follow a specific binomial distribution, *i.e.*, $\mathbf{A}^{(m),+} \propto \binom{n}{m}, \mathbf{A}^{(m),-} \propto \binom{n}{m}$, where $\binom{n}{m} = \frac{n!}{m!(n-m)!}$ represents the number of combinations of $m$ input variables out of $n$ input variables (see Appendix E for the proof). We notice that not all input variables interact with each other. Thus, we introduce a scaling coefficient $\alpha$ to obtain the analytic distribution of non-generalizable interactions, denoted by $\boldsymbol{\mathcal{A}}_{\text{spindle}}^{(m),+}$ and $\boldsymbol{\mathcal{A}}_{\text{spindle}}^{(m),-}$:

$$\boldsymbol{\mathcal{A}}_{\text{spindle}}^{(m),+}(\alpha, \beta) = -\boldsymbol{\mathcal{A}}_{\text{spindle}}^{(m),-}(\alpha, \beta) = \beta \cdot \frac{\Gamma(\alpha n + 1)}{\Gamma(m+1) \cdot \Gamma(\alpha n - m + 1)}. \tag{7}$$

Here, $\Gamma(\cdot)$ denotes the gamma function, which extends the binomial distribution $\binom{\alpha n}{m}$ to the domain of real numbers, *s.t.* $\Gamma(k) = (k-1)!$. The coefficient $\beta$ scales the interactions magnitude.

**Despite relying on some simplified assumptions, the estimated distribution accurately predicts the real non-generalizable interactions in later experiments (see Figure 4 and Appendix G).**

### 3.2 Modeling the decay-shaped distribution

In this subsection, we aim to formulate the distribution of generalizable interactions. However, there is no direct theoretical framework to describe or constrain the generalizability of interactions. Fortunately, Ren et al. (2024b) have proven that introducing parameter perturbations during the training process typically encourages the DNN to model more low-order interactions.

Since training under parameter perturbation has been widely recognized as a typical method to enhance model generalizability (Achille & Soatto, 2018), we formulate the distribution of generalizable interactions based on (Ren et al., 2024b). Although our theoretical derivation relies on several simplified assumptions, subsequent experiments show that our theory successfully decomposes the generalizable interactions in a DNN.

**Theorem 3.1.** *(Proved by Ren et al. (2024b)[6]) Let $\boldsymbol{I}^* = \left[I_{S_1}^*, I_{S_2}^*, \ldots, I_{S_{2^n}}^*\right]^T$ denote the effects of all $2^n$ interactions extracted from a fully converged DNN, which is likely overfitted. Then, if we inject uncertainty/noises of magnitude $\delta$ into the intermediate-layer features of the DNN and finetune the DNN, then the DNN would reduce high-order interactions, yielding the following set of interactions:*

$$\hat{\boldsymbol{I}}(\delta) = M(\delta)\, \boldsymbol{I}^*, \tag{8}$$

*where $M(\delta) \in \mathbb{R}^{2^n \times 2^n}$ is a matrix. The detailed formulation of $M$ is provided in Appendix I due to the page limit. A larger value of $\delta$ penalizes the strength of high-order interactions more significantly.*

Theorem 3.1 clarifies the explicit effects on the distribution of interactions when the DNN is trained by injecting noises into intermediate features. According to (Achille & Soatto, 2018), this approach is commonly employed to boost the generalization power of a DNN. It eliminates counteracting high-order interactions and pushes the distribution towards a decay shape. Based on this, we formulate the strength of positive interaction effect and that of negative interaction effect of each $m$-th order, denoted by $\boldsymbol{\mathcal{A}}_{\text{decay}}^{(m),+}$ and $\boldsymbol{\mathcal{A}}_{\text{decay}}^{(m),-}$, as follows:

$$\begin{aligned}
\boldsymbol{\mathcal{A}}_{\text{decay}}^{(m),+}(\delta, \gamma) &= \gamma \sum\nolimits_{S:|S|=m} \max\{\hat{I}_S^{\text{AND}}(\delta), 0\} + \max\{\hat{I}_S^{\text{OR}}(\delta), 0\}, \\
\boldsymbol{\mathcal{A}}_{\text{decay}}^{(m),-}(\delta, \gamma) &= \gamma \sum\nolimits_{S:|S|=m} \min\{\hat{I}_S^{\text{AND}}(\delta), 0\} + \min\{\hat{I}_S^{\text{OR}}(\delta), 0\},
\end{aligned} \tag{9}$$

where coefficient $\gamma$ scales the magnitude of interactions.

### 3.3 Disentangling the generalizable and non-generalizable interactions

We propose a method to disentangle the two distributions of interactions from a given DNN. Although the distributions in Equation (7) and Equation (9) are derived based on some simplifying assumptions and heuristic settings, experiments in Section 3.4 verify the effectiveness of our theory.

---

[6]Although the theorem appear complex, experiments in (Ren et al., 2024b) and Appendix I empirically validate its ability to accurately predict the dynamics of interaction distribution when the DNN is not overfitted.

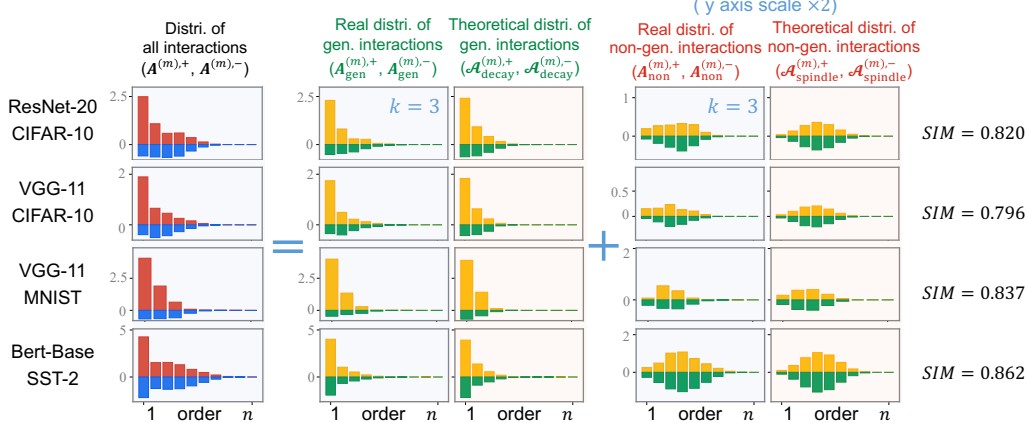

Figure 4: Verification of our proposed method's accuracy in capturing the true distributions of generalizable interactions and non-generalizable interactions. Thedemonstrates a high degree of alignment between the theoretical disentangled distributions (represented by $\mathcal{A}_{\text{decay}}^{(m),+}$, $\mathcal{A}_{\text{decay}}^{(m),-}$, $\mathcal{A}_{\text{spindle}}^{(m),+}$, $\mathcal{A}_{\text{spindle}}^{(m),-}$) and the real distributions (represented by $\mathbf{A}_{\text{gen}}^{(m),+}$, $\mathbf{A}_{\text{gen}}^{(m),-}$, $\mathbf{A}_{\text{non}}^{(m),+}$, $\mathbf{A}_{\text{non}}^{(m),-}$), as evidenced by the consistently high Jaccard similarity values ($SIM$). To enhance clarity, we double the y-axis scale for non-generalizable interaction distributions. **More results are provided in Appendix L.3.**

Given a DNN, let $\mathbf{A}^{(m),+}$ and $\mathbf{A}^{(m),-}$ represent the total strength of $m$-th order positive and negative interactions, respectively. The distributions of generalizable interactions ($\mathcal{A}_{\text{decay}}^{(m),+}$, $\mathcal{A}_{\text{decay}}^{(m),-}$) and non-generalizable interactions ($\mathcal{A}_{\text{spindle}}^{(m),+}$, $\mathcal{A}_{\text{spindle}}^{(m),-}$) can be disentangled by optimizing the following parameter model:

$$\min_{\alpha,\beta;\delta,\gamma} \sum_{m=1}^{n} (\mathbf{A}^{(m),+} - \mathcal{A}_{\text{theory}}^{(m),+})^2 + (\mathbf{A}^{(m),-} - \mathcal{A}_{\text{theory}}^{(m),-})^2$$

$$\text{s.t.} \quad \mathcal{A}_{\text{theory}}^{(m),+} = \mathcal{A}_{\text{decay}}^{(m),+}(\delta,\gamma) + \mathcal{A}_{\text{spindle}}^{(m),+}(\alpha,\beta), \; \mathcal{A}_{\text{theory}}^{(m),-} = \mathcal{A}_{\text{decay}}^{(m),-}(\delta,\gamma) + \mathcal{A}_{\text{spindle}}^{(m),-}(\alpha,\beta). \tag{10}$$

### 3.4 EXPERIMENTAL VERIFICATION

**Faithfulness of the disentangled generalizable and non-generalizable interactions.** We conducted experiments to validate whether the proposed method in Section 3.3 accurately captures the true distributions of generalizable and non-generalizable interactions. Specifically, we compared the theoretically disentangled distributions of decay-shaped distribution and spindle-shaped distribution (represented by $\mathcal{A}_{\text{decay}}^{(m),+}$, $\mathcal{A}_{\text{decay}}^{(m),-}$, $\mathcal{A}_{\text{spindle}}^{(m),+}$, $\mathcal{A}_{\text{spindle}}^{(m),-}$) with the real distributions of generalizable and non-generalizable interactions (represented by $\mathbf{A}_{\text{gen}}^{(m),+}$, $\mathbf{A}_{\text{gen}}^{(m),-}$, $\mathbf{A}_{\text{non}}^{(m),+}$, $\mathbf{A}_{\text{non}}^{(m),-}$), respectively. The matching quality was evaluated using the Jaccard similarity as follows:

$$SIM = \frac{\sum_{m=1}^{n} \sum_{s\in\{+,-\}} \min\{|\mathbf{A}_{\text{gen}}^{(m),s}|, |\mathcal{A}_{\text{decay}}^{(m),s}|\} + \min\{|\mathbf{A}_{\text{non}}^{(m),s}|, |\mathcal{A}_{\text{spindle}}^{(m),s}|\}}{\sum_{m=1}^{n} \sum_{s\in\{+,-\}} \max\{|\mathbf{A}_{\text{gen}}^{(m),s}|, |\mathcal{A}_{\text{decay}}^{(m),s}|\} + \max\{|\mathbf{A}_{\text{non}}^{(m),s}|, |\mathcal{A}_{\text{spindle}}^{(m),s}|\}}. \tag{11}$$

*Figure 4 shows consistently high Jaccard similarity values ($SIM$), indicating that our method accurately captures the real interaction distributions.*

Additionally, we observed that the distribution of generalizable interactions measured in baseline DNNs exhibited a few slightly higher-order terms compared to the theoretically predicted decay-shaped distribution. This minor discrepancy may arise from the inherent challenge of entirely preventing baseline DNNs from learning noise patterns. There remained a small but non-zero probability that noise-induced interactions could transfer from target DNNs to the baseline network.

**Using the spindle-shaped distribution to quantify noise representations.** This experiment evaluated the faithfulness of disentangled spindle-shaped distribution from a new perspective. As a supplement to the experiments in Section 2.3, which used transferability to baseline DNNs to construct ground-truth non-generalizable interactions, we here established non-generalizable interactions by injecting noise to a well-trained DNN. Specifically, we considered two noise types: (1) Adding Gaussian noises $\epsilon \sim \mathcal{N}(0, \sigma^2)$ to the network parameters; and (2) Injecting adversarial perturbations[7]

---

[7]The Fast Gradient Sign Method (FGSM) (Goodfellow, 2014) was used to generate adversarial perturbations: $\boldsymbol{x}_{\text{noise}} = \boldsymbol{x} + \sigma \cdot \text{sign}(\nabla_{\boldsymbol{x}} v(\boldsymbol{x}))$, where $\sigma$ is the perturbation magnitude.

(a) Tracking the change of two distributions when we injected more non-generalizable representations to DNN's parameters.

(b) Tracking the change of two distributions when we injected more non-generalizable representations to input samples.

Interactions in the decay-shaped distribution remained largely unaffected by $\sigma^2$

Interactions in the spindle-shaped distribution were enhanced by increase $\sigma^2$

The matching error was small

Figure 5: As we increase the noise magnitudes $\sigma^2$, *i.e.*, injecting more non-generalizable representations into the DNN, the significance of interactions in the spindle-shaped distribution ($\mathcal{A}_{\text{spindle}}^{(m),+}$ and $\mathcal{A}_{\text{spindle}}^{(m),-}$) increases. In contrast, interactions in the decay-shaped distribution ($\mathcal{A}_{\text{decay}}^{(m),+}$ and $\mathcal{A}_{\text{decay}}^{(m),-}$) remain largely unaffected. This observation further validates the faithfulness of our method. **Additional results for varying values of $\sigma$ and more DNNs are provided in Appendix L.4.**

into input sample $x$ to generate $x_{\text{noise}}$. For both scenarios, we computed the interaction distributions $(\mathbf{A}^{(m),+}, \mathbf{A}^{(m),-})$. Then, we utilized the disentangling methods following Equation (10) to extract decay-shaped distribution ($\mathcal{A}_{\text{decay}}^{(m),+}, \mathcal{A}_{\text{decay}}^{(m),-}$) and spindle-shaped distribution ($\mathcal{A}_{\text{spindle}}^{(m),+}, \mathcal{A}_{\text{spindle}}^{(m),-}$).

We trained ResNet-20 and VGG-11 on the MNIST dataset and introduced non-generalizable representations of varying magnitudes into the DNNs by progressively increasing the values of $\sigma^2$. As shown in Figure 5, the disentangled spindle-shaped distribution ($\mathcal{A}_{\text{spindle}}^{(m),+}, \mathcal{A}_{\text{spindle}}^{(m),-}$) successfully captured the increasing non-generalizable representations along with the increasing noise magnitude $\sigma^2$. In contrast, the disentangled decay-shaped distribution of generalizable interactions was not significantly affected. Moreover, the matching error ($\mathcal{A}_{\text{residual}}^{(m),+} = |\mathcal{A}_{\text{theory}}^{(m),+} - \mathbf{A}^{(m),+}|$, $\mathcal{A}_{\text{residual}}^{(m),-} = |\mathcal{A}_{\text{theory}}^{(m),-} - \mathbf{A}^{(m),-}|$) was small. In this way, this experiment also verified the effectiveness of our theory. *All the above experimental results demonstrate that the proposed metrics ($\mathcal{A}_{\text{decay}}^{(m),+}, \mathcal{A}_{\text{decay}}^{(m),-}$) and ($\mathcal{A}_{\text{spindle}}^{(m),+}, \mathcal{A}_{\text{spindle}}^{(m),-}$) unambiguously distinguish between generalizable and non-generalizable interactions.*

# 4 CONCLUSION AND DISCUSSIONS

In this paper, we analyze the generalization power of a DNN in terms of the generalization power of interactions encoded by the DNN. We discover and further experimentally verify that generalizable interactions in a DNN follow a decay-shaped distribution across different orders, whereas non-generalizable interactions exhibit a spindle-shaped distribution. We further propose a method to disentangle these two distributions within a given DNN. Extensive experimental results validate that our theory accurately captures the true distributions of interactions.

**Theoretical contributions and limitations.** Our method provides a new perspective that extends the traditional end-to-end testing paradigm in deep learning. Specifically, it allows us to evaluate DNN representation quality without requiring numerous testing data. However, our work is still not an ultimate explanation of the generalization power of interactions. Our findings primarily describe the general distribution trends of generalizable and non-generalizable interactions across different orders. Determining the generalizability of each individual interaction still requires the baseline DNN.

**Application potential.** Our theory shows its potential in several applications. First, as shown in follow-up studies (see Appendix J for the details), penalizing non-generalizable interactions during training would substantially improve the DNN's performance. Second, our approach explains model representation quality at a mechanistic level without requiring extensive test samples. This facilitates the direct identification of representation errors without relying on end-to-end testing paradigms.

Particularly, our theory may help determine the optimal early stopping point during DNN training by tracking the generalizability of interactions encoded by the DNN. In a preliminary experiment, we have disentangled the distribution of generalizable interactions ($\mathcal{A}_{\text{decay}}^{(m),+}$ and $\mathcal{A}_{\text{decay}}^{(m),-}$) and that of non-generalizable interactions ($\mathcal{A}_{\text{spindle}}^{(m),+}$ and $\mathcal{A}_{\text{spindle}}^{(m),-}$) during the training process. Typically, as shown in Figure 3, the strengthening of non-generalizable interactions in the spindle-shaped distribution is a clear sign of overfitting.

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

# A    RELATED WORK

Analyzing and quantifying the generalization power of deep neural networks (DNNs) is a fundamental challenge in deep learning research. Existing approaches primarily focus on two perspectives: (1) analyzing the loss gapNeyshabur et al. (2017); Bousquet et al. (2020); Deng et al. (2021); Haghifam et al. (2020; 2021), and (2) examining the smoothness of the loss landscapeKeskar et al. (2016); Li et al. (2018); Foret et al. (2021); Kwon et al. (2021). Additionally, some studies investigate the generalization power of DNNs in high-dimensional feature spaces (Petrini et al., 2022; Boopathy et al., 2023; Dyballa et al., 2024; Nikolikj et al., 2024).

However, recent advances in interaction-based theory provide a novel and direct perspective for analyzing the generalization power of DNNs. Specifically, interaction-based methods define and quantify the interaction effects encoded by DNNs. Since the output of a DNN can be faithfully decomposed into the sum of all AND-OR interactions, the generalization power of the DNN can be viewed as the aggregation of the generalization power of these interactions.

**Literature in guaranteeing the faithfulness of defining and disentangling a DNN's inference patterns.** Ren et al. (2023a) discovered, and Ren et al. (2024a) later proved, that given a DNN, there always exists a logical model consisting of AND-OR inference patterns that can accurately predict the DNN's outputs on all $2^n$ masked states of an input sample using a small set of AND-OR interactions. Furthermore, Li & Zhang (2023) demonstrated that salient interactions extracted from a DNN are shared across different samples and exhibit remarkable discriminative power in classification tasks. Chen et al. (2024) proposed a method to extract interactions that are generalizable across different models. These findings suggest that interactions serve as primitive inference patterns encoded by DNNs, forming the theoretical foundation for interaction-based frameworks.

**Literature in explaining the generalization power of DNNs from the perspective of interactions.** Recent research highlights the critical role of interactions in explaining hidden factors that influence a DNN's adversarial robustnessRen et al. (2021), adversarial transferabilityWang et al. (2021), and generalization powerZhou et al. (2024). For instance, Deng et al. (2022) discovered and theoretically proved the existence of a representation bottleneck in DNNs, which limits their ability to encode interactions of intermediate complexity. Ren et al. (2023b) found that Bayesian neural networks (BNNs) tend to avoid encoding complex interactions, providing an explanation for the strong adversarial robustness of BNNs. Liu et al. (2023) demonstrated that DNNs learn simple interactions more easily than complex ones. Zhang et al. (2024) and Ren et al. (2024b) analyzed the two-phase dynamics in the learning process of DNNs, further elucidating the role of interactions in generalization.

# B    EXAMPLES OF AND-OR LOGICAL MODELS FOR INTERPRETING LLM OUTPUTS

This section provides additional examples of AND-OR logical models that interpret the outputs of two language models—DeepSeekR1-Distill-Llama-8B and Qwen2.5-7B—across diverse input prompts. As illustrated in Figure 6, the proposed logical framework consistently explains model predictions even when input tokens are partially masked at random.

**Prompt 1:**    New York Department of Health recommends that all people should wear N95, KN95, or KF94 masks in all public

**Target:**    settings

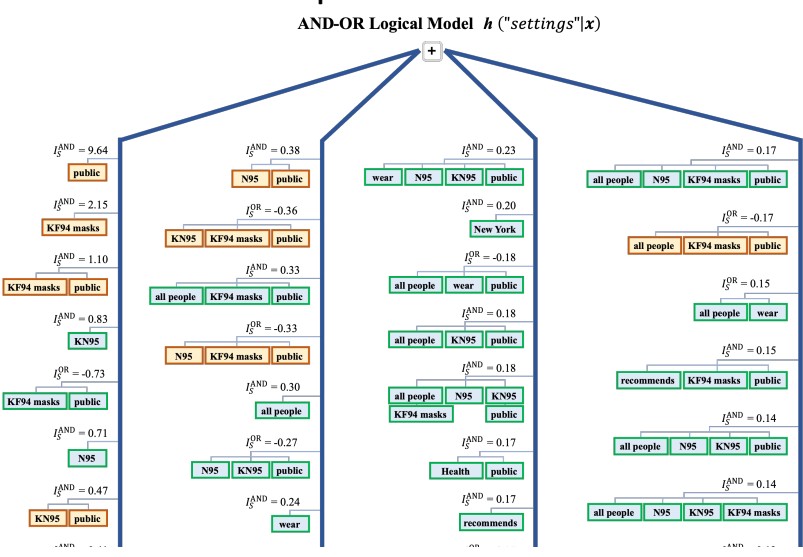

**Prompt 2:**    On June 11, 2018, OpenAI researchers and engineers published a paper introducing the first generative pre-trained

**Target:**    transformer

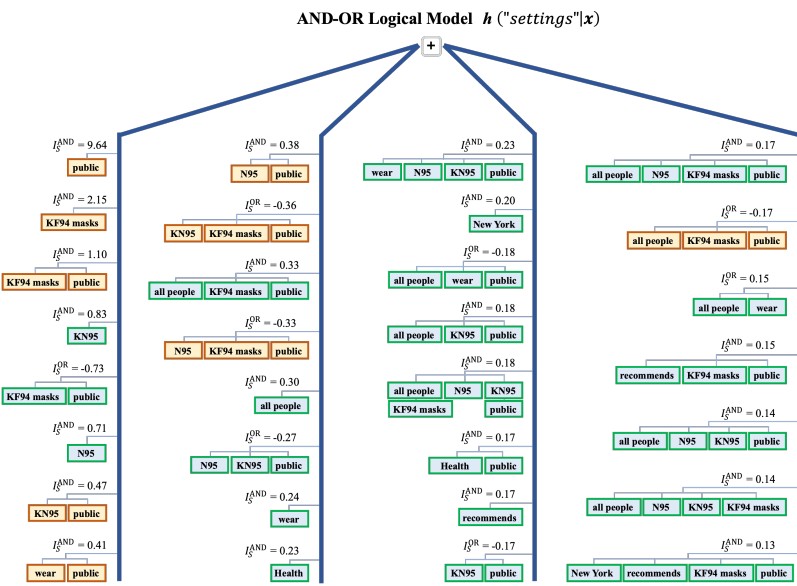

**Prompt 1:** New York Department of Health recommends that all people should wear N95, KN95, or KF94 masks in all public

**Target:** settings

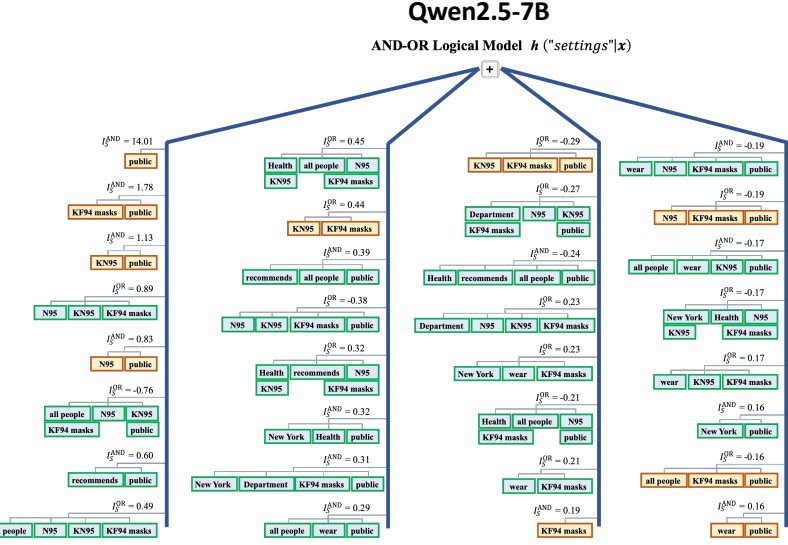

**Prompt 2:** On June 11, 2018, OpenAI researchers and engineers published a paper introducing the first generative pre-trained

**Target:** transformer

Figure 6: Examples of AND-OR logical models interpreting LLM outputs under arbitrary token masking (Part 1). Two prompt-completion pairs are shown: (1) **Prompt**: "New York Department of Health recommends that all people should wear N95, KN95, or KF94 masks in all public". **Generated output**: "settings". (2) **Prompt**: "On June 11, 2018, OpenAI researchers and engineers published a paper introducing the first generative pre-trained". **Generated output**: "transformer". The logical model identifies necessary and sufficient conditions that lead to the model's prediction, even when parts of the input are occluded.

## C THREE COMMON CONDITIONS FOR SPARSE INTERACTIONS

Ren et al. (2024a) have formulated three mathematical conditions for the sparsity of AND interactions, as follows.

**Condition 1.** *The DNN does not encode interactions higher than the $M$-th order:* $\forall S \in \{S \subseteq N \mid |S| \geq M+1\}, I_S^{AND}.$

Condition 1 implies that the DNN does not encode extremely high-order interactions. This is because extremely high-order interactions usually represent very complex and over-fitted patterns, which are unnecessary and unlikely to be learned by the DNN in real applications.

**Condition 2.** *Let us consider the average network output $\bar{u}^{(k)} \overset{\text{def}}{=} \mathbb{E}_{|S|=k}[v(\boldsymbol{x}_S) - v(\boldsymbol{x}_\emptyset)]$ over all masked samples $\boldsymbol{x}_S$ with $k$ unmasked input variables. This average network output monotonically increases when $k$ increases: $\forall k' \leq k$, we have $\bar{u}^{(k')} \leq \bar{u}^{(k)}$.*

Condition 2 implies that a well-trained DNN is likely to have higher classification confidence for input samples that are less masked.

**Condition 3.** *Given the average network output $\bar{u}^{(k)}$ of samples with $k$ unmasked input variables, there is a polynomial lower bound for the average network output of samples with $k'(k' \leq k)$ unmasked input variables: $\forall k' \leq k, \bar{u}^{(k')} \geq (\frac{k'}{k})^p \bar{u}^{(k)}$, where $p > 0$ is a positive constant.*

Condition 3 implies that the classification confidence of the DNN does not significantly degrade on masked input samples. The classification/detection of masked/occluded samples is common in real scenarios. In this way, a well-trained DNN usually learns to classify a masked input sample based on local information (which can be extracted from unmasked parts of the input) and thus should not yield a significantly low confidence score on masked samples.

## D PROOF OF UNIVERSAL MATCHING PROPERTY (THEOREM 2.1)

*Proof.* **(1) Universal matching property of AND interactions.**

We prove that the output component $u_{\text{and}}(\boldsymbol{x}_S)$ on all $2^n$ masked samples $\{\boldsymbol{x}_S : S \subseteq N\}$ could be universally explained by the all interactions in $S \subseteq N$, *i.e.*, $\forall \emptyset \neq S \subseteq N, u_{\text{and}}(\boldsymbol{x}_S) = \sum_{\emptyset \neq T \subseteq S} I_T^{\text{AND}} + u(\boldsymbol{x}_\emptyset)$. In particular, we define $u_{\text{and}}(\boldsymbol{x}_\emptyset) = v(\boldsymbol{x}_\emptyset)$ (*i.e.*, we attribute output on an empty sample to AND interactions).

Specifically, the AND interaction is defined as $I_T^{\text{AND}} = \sum_{L \subseteq T}(-1)^{|T|-|L|} u_L^{\text{AND}}$ in Theorem 2.1 To compute the sum of AND interactions $\sum_{\emptyset \neq T \subseteq S} I_T^{\text{AND}} = \sum_{\emptyset \neq T \subseteq S} \sum_{L \subseteq T}(-1)^{|T|-|L|} u_L^{\text{AND}}$, we first exchange the order of summation of the set $L \subseteq T \subseteq S$ and the set $T \supseteq L$. That is, we compute all linear combinations of all sets $T$ containing $L$ with respect to the model outputs $u_L^{\text{AND}}$ given a set of input variables $L$, *i.e.*, $\sum_{T:L \subseteq T \subseteq S}(-1)^{|T|-|L|} u_L^{\text{AND}}$. Then, we compute all summations over the set $L \subseteq S$.

In this way, we can compute them separately for different cases of $L \subseteq T \subseteq S$. In the following, we consider the cases (1) $L = S = T$, and (2) $L \subseteq T \subseteq S, L \neq S$, respectively.

**Case 1:** $L = S = T$, the linear combination of all subsets $T$ containing $L$ with respect to the model output $u_L^{\text{AND}}$ is $(-1)^{|S|-|S|} u_L^{\text{AND}} = u_L^{\text{AND}}$.

**Case 2:** $L \subseteq T \subseteq S, L \neq S$, the linear combination of all subsets $T$ containing $L$ with respect to the model output $u_L^{\text{AND}}$ is $\sum_{T:L \subseteq T \subseteq S}(-1)^{|T|-|L|} u_L^{\text{AND}}$. For all sets $T : S \supseteq T \supseteq L$, let us consider the linear combinations of all sets $T$ with number $|T|$ for the model output $u_L^{\text{AND}}$, respectively. Let $m := |T| - |L|, (0 \leq m \leq |S| - |L|)$, then there are a total of $C_{|S|-|L|}^m$ combinations of all sets $T$ of order $|T|$. Thus, given $L$, accumulating the model outputs $u_L^{\text{AND}}$ corresponding to all $T \supseteq L$, then

$$\sum_{T:L \subseteq T \subseteq S}(-1)^{|T|-|L|} u_L^{\text{AND}} = u_L^{\text{AND}} \cdot \underbrace{\sum_{m=0}^{|S|-|L|} C_{|S|-|L|}^m (-1)^m}_{=0} = 0. \text{ Please see the complete}$$

derivation of the following formula.

$$\sum_{\emptyset \neq T \subseteq S} I_{\text{and}}(T|\boldsymbol{x})$$

$$= \sum_{\emptyset \neq T \subseteq S} \sum_{L \subseteq T} (-1)^{|T|-|L|} u_L^{\text{AND}}$$

$$= \sum_{L \subseteq S} \sum_{T:L \subseteq T \subseteq S} (-1)^{|T|-|L|} u_L^{\text{AND}} - u_{\emptyset}^{\text{AND}}$$

$$= \underbrace{u_S^{\text{AND}}}_{L=S} + \sum_{L \subseteq S, L \neq S} u_L^{\text{AND}} \cdot \underbrace{\sum_{m=0}^{|S|-|L|} C_{|S|-|L|}^m (-1)^m}_{=0} - u_{\emptyset}^{\text{AND}} \qquad (12)$$

$$= u_S^{\text{AND}} - u_{\emptyset}^{\text{AND}}$$

$$= u_S^{\text{AND}} - u_{\emptyset}$$

Thus, we have $\forall \emptyset \neq S \subseteq N, u_{\text{and}}(\boldsymbol{x}_S) = \sum_{\emptyset \neq T \subseteq S} I_T^{\text{AND}} + v(\boldsymbol{x}_{\emptyset})$.

**(2) Universal matching property of OR interactions.**

According to the definition of OR interactions, we will derive that $\forall S \subseteq N, u_S^{\text{OR}} = \sum_{T:T \cap S \neq \emptyset} I_S^{\text{OR}}$, where we define $u_{\emptyset}^{\text{OR}} = 0$ (recall that in Step (1), we attribute the output on empty input to AND interactions).

Specifically, the OR interaction is defined as $I_S^{\text{OR}} = -\sum_{L \subseteq T} (-1)^{|T|-|L|} u_{N \setminus L}^{\text{OR}}$ in Theorem 2.1 Similar to the above derivation of the Universal matching property of AND interactions, to compute the sum of OR interactions $\sum_{T:T \cap S \neq \emptyset} I_T^{\text{OR}} = \sum_{T:T \cap S \neq \emptyset} \left[ -\sum_{L \subseteq T} (-1)^{|T|-|L|} u_{N \setminus L}^{\text{OR}} \right]$, we first exchange the order of summation of the set $L \subseteq T \subseteq N$ and the set $T : T \cap S \neq \emptyset$. That is, we compute all linear combinations of all sets $T$ containing $L$ with respect to the model outputs $u_{N \setminus L}^{\text{OR}}$ given a set of input variables $L$, *i.e.*, $\sum_{T:T \cap S \neq \emptyset, T \supseteq L} (-1)^{|T|-|L|} u_{N \setminus L}^{\text{OR}}$. Then, we compute all summations over the set $L \subseteq N$.

In this way, we can compute them separately for different cases of $L \subseteq T \subseteq N, T \cap S \neq \emptyset$. In the following, we consider the cases (1) $L = N \setminus S$, (2) $L = N$, (3) $L \cap S \neq \emptyset, L \neq N$, and (4) $L \cap S = \emptyset, L \neq N \setminus S$, respectively.

**Case 1:** $L = N \setminus S$, the linear combination of all subsets $T$ containing $L$ with respect to the model output $u_{N \setminus L}^{\text{OR}}$ is $\sum_{T:T \cap S \neq \emptyset, T \supseteq L} (-1)^{|T|-|L|} u_{N \setminus L}^{\text{OR}} = \sum_{T:T \cap S \neq \emptyset, T \supseteq L} (-1)^{|T|-|L|} u_S^{\text{OR}}$. For all sets $T : T \supseteq L, T \cap S \neq \emptyset$ (then $T \neq N \setminus S, T \neq L$), let us consider the linear combinations of all sets $T$ with number $|T|$ for the model output $u_S^{\text{OR}}$, respectively. Let $|T'| := |T| - |L|, (1 \leq |T'| \leq |S|)$, then there are a total of $C_{|S|}^{|T'|}$ combinations of all sets $T'$ of order $|T'|$. Thus, given $L$, accumulating the model outputs $u_S^{\text{OR}}$ corresponding to all $T \supseteq L$, then $\sum_{T:T \cap S \neq \emptyset, T \supseteq L} (-1)^{|T|-|L|} u_{N \setminus L}^{\text{OR}} = u_S^{\text{OR}} \cdot \underbrace{\sum_{|T'|=1}^{|S|} C_{|S|}^{|T'|} (-1)^{|T'|}}_{=-1} = -u_S^{\text{OR}}$.

**Case 2:** $L = N$ (then $T = N$), the linear combination of all subsets $T$ containing $L$ with respect to the model output $u_{N \setminus L}^{\text{OR}}$ is $\sum_{T:T \cap S \neq \emptyset, T \supseteq L} (-1)^{|T|-|L|} u_{N \setminus L}^{\text{OR}} = (-1)^{|N|-|N|} u_{\emptyset}^{\text{OR}} = u_{\emptyset}^{\text{OR}}$.

**Case 3:** $L \cap S \neq \emptyset, L \neq N$, the linear combination of all subsets $T$ containing $L$ with respect to the model output $u_{N \setminus L}^{\text{OR}}$ is $\sum_{T:T \cap S \neq \emptyset, T \supseteq L} (-1)^{|T|-|L|} u_{N \setminus L}^{\text{OR}}$. For all sets $T : T \supseteq L, T \cap S \neq \emptyset$, let us consider the linear combinations of all sets $T$ with number $|T|$ for the model output $u_S^{\text{OR}}$, respectively. Let us split $|T| - |L|$ into $|T'|$ and $|T''|$, *i.e.,* $|T| - |L| = |T'| + |T''|$, where $T' = \{i | i \in T, i \notin L, i \in N \setminus S\}$, $T'' = \{i | i \in T, i \notin L, i \in S\}$ (then $0 \leq |T''| \leq |S| - |S \cap L|$) and $|T'| + |T''| + |L| = |T|$. In this way, there are a total of $C_{|S|-|S \cap L|}^{|T''|}$ combinations of all sets $T''$ of order $|T''|$. Thus, given $L$, accumulating the model outputs $u_{\text{OR}}(\boldsymbol{x}_{N \setminus L})$ corresponding to all $T \supseteq L$, then

$$\sum_{T:T \cap S \neq \emptyset, T \supseteq L} (-1)^{|T|-|L|} u_{N \setminus L}^{\text{OR}} = u_{N \setminus L}^{\text{OR}} \sum_{T' \subseteq N \setminus S \setminus L} \underbrace{\sum_{|T''|=0}^{|S|-|S \cap L|} C_{|S|-|S \cap L|}^{|T''|} (-1)^{|T'|+|T''|}}_{=0} = 0.$$

**Case 4:** $L \cap S = \emptyset, L \neq N \setminus S$, the linear combination of all subsets $T$ containing $L$ with respect to the model output $u_{N \setminus L}^{\text{OR}}$ is $\sum_{T:T \cap S \neq \emptyset, T \supseteq L}(-1)^{|T|-|L|}u_{N \setminus L}^{\text{OR}}$. Similarly, let us split $|T| - |L|$ into $|T'|$ and $|T''|$, $i.e.,|T| - |L| = |T'| + |T''|$, where $T' = \{i|i \in T, i \notin L, i \in N \setminus S\}$, $T'' = \{i|i \in T, i \in S\}$ (then $0 \leq |T''| \leq |S|$) and $|T'| + |T''| + |L| = |T|$. In this way, there are a total of $C_{|S|}^{|T''|}$ combinations of all sets $T''$ of order $|T''|$. Thus, given $L$, accumulating the model outputs $u_{\text{OR}}(\boldsymbol{x}_{N \setminus L})$ corresponding to all $T \supseteq L$, then $\sum_{T:T \cap S \neq \emptyset, T \supseteq L}(-1)^{|T|-|L|}u_{N \setminus L}^{\text{OR}} = u_{N \setminus L}^{\text{OR}} \cdot \sum_{T' \subseteq N \setminus S \setminus L} \underbrace{\sum_{|T''|=0}^{|S|} C_{|S|}^{|T''|}(-1)^{|T'|+|T''|}}_{=0} = 0.$

Please see the complete derivation of the following formula.

$$
\begin{aligned}
\sum_{T:T \cap S \neq \emptyset} I_T^{\text{OR}} &= \sum_{T:T \cap S \neq \emptyset}\left[-\sum_{L \subseteq T}(-1)^{|T|-|L|}u_{N \setminus L}^{\text{OR}}\right]\\
&= -\sum_{L \subseteq N}\sum_{T:T \cap S \neq \emptyset, T \supseteq L}(-1)^{|T|-|L|}u_{N \setminus L}^{\text{OR}}\\
&= -\left[\sum_{|T'|=1}^{|S|}C_{|S|}^{|T'|}(-1)^{|T'|}\right]\cdot \underbrace{u_S^{\text{OR}}}_{L=N \setminus S} - \underbrace{u_{\emptyset}^{\text{OR}}}_{L=N}\\
&\quad -\sum_{L \cap S \neq \emptyset, L \neq N}\left[\sum_{T' \subseteq N \setminus S \setminus L}\left(\sum_{|T''|=0}^{|S|-|S \cap L|}C_{|S|-|S \cap L|}^{|T''|}(-1)^{|T'|+|T''|}\right)\right]\cdot u_{N \setminus L}^{\text{OR}}\\
&\quad -\sum_{L \cap S = \emptyset, L \neq N \setminus S}\left[\sum_{T' \subseteq N \setminus S \setminus L}\left(\sum_{|T''|=0}^{|S|}C_{|S|}^{|T''|}(-1)^{|T'|+|T''|}\right)\right]\cdot u_{N \setminus L}^{\text{OR}} \qquad (13)\\
&= -(-1)\cdot u_S^{\text{OR}} - u_{\emptyset}^{\text{OR}} - \sum_{L \cap S \neq \emptyset, L \neq N}\left[\sum_{T' \subseteq N \setminus S \setminus L}0\right]\cdot u_{N \setminus L}^{\text{OR}}\\
&\quad -\sum_{L \cap S = \emptyset, L \neq N \setminus S}\left[\sum_{T' \subseteq N \setminus S \setminus L}0\right]\cdot u_{N \setminus L}^{\text{OR}}\\
&= u_S^{\text{OR}} - u_{\emptyset}^{\text{OR}}\\
&= u_S^{\text{OR}}
\end{aligned}
$$

**(3) Universal matching property of AND-OR interactions.**

With the Universal matching property of AND interactions and the Universal matching property of OR interactions, we can easily get $v(\boldsymbol{x}_S) = u_S^{\text{AND}} + u_S^{\text{OR}} = v(\boldsymbol{x}_{\emptyset}) + \sum_{\emptyset \neq T \subseteq S}I_T^{\text{AND}} + \sum_{T:T \cap S \neq \emptyset}I_T^{\text{OR}}$, thus, we obtain the Universal matching property of AND-OR interactions.

$\square$

# E  PROOF OF SPINDLE-SHAPED DISTRIBUTION OF NON-GENERALIZABLE INTERACTIONS IN SECTION 3.1

**Theorem E.1.** *Under the assumption that each interaction in a randomly initialized deep neural network (DNN) follows a Gaussian distribution $I_S \sim \mathcal{N}(0, \sigma^2)$ (Zhang et al., 2024), the distribution of non-generalizable $m$-order interactions is characterized by a binomial distribution. Specifically, the sum effects of positive $m$-order interactionssand negative $m$-order interactions, denoted as $\boldsymbol{A}^{(m),+}$ and $\boldsymbol{A}^{(m),-}$, respectively, follow $\boldsymbol{A}^{(m),+} \sim \binom{n}{m}$ and $\boldsymbol{A}^{(m),-} \sim \binom{n}{m}$.*

**Lemma E.2.** *Let $X_1, X_2, \ldots, X_n$ be standard Gaussian random variables independent and identically distributed (i.i.d.), $X_i \sim \mathcal{N}(0, 1)$. Define the sum of positive and negative samples as:*

$$
S_+ = \sum_{i=1}^n X_i \cdot \mathbb{I}(X_i > 0), \quad S_- = \sum_{i=1}^n X_i \cdot \mathbb{I}(X_i < 0),
$$

*where $\mathbb{I}(\cdot)$ is the indicator function.*

*(1) Means of $S_+$ and $S_-$* *The expected value of $S_+$ is given by:*

$$\mathbb{E}[S_+] = n \cdot \mathbb{E}[X_i \cdot \mathbb{I}(X_i > 0)] = n \cdot \int_0^\infty x \cdot f_X(x)\, dx,$$

*where $f_X(x)$ is the probability density function (PDF) of $\mathcal{N}(0,1)$. Due to the symmetry of $X_i$ about zero, we have:*

$$\mathbb{E}[X_i \cdot \mathbb{I}(X_i > 0)] = \frac{1}{2}\mathbb{E}[|X_i|] = \frac{1}{2} \cdot \sqrt{\frac{2}{\pi}} = \frac{1}{\sqrt{2\pi}}.$$

*Thus, the means of $S_+$ and $S_-$ are:*

$$\mathbb{E}[S_+] = n \cdot \frac{1}{\sqrt{2\pi}}, \quad \mathbb{E}[S_-] = -n \cdot \frac{1}{\sqrt{2\pi}}.$$

*(2) Variances of $S_+$ and $S_-$* *The variances of $S_+$ and $S_-$ are computed as:*

$$Var(S_+) = Var(S_-) = n \cdot Var(X_i \cdot \mathbb{I}(X_i > 0)) = n \cdot \left( \mathbb{E}[X_i^2 \cdot \mathbb{I}(X_i > 0)] - (\mathbb{E}[X_i \cdot \mathbb{I}(X_i > 0)])^2 \right).$$

*Since $\mathbb{E}[X_i^2 \cdot \mathbb{I}(X_i > 0)] = \frac{1}{2}\mathbb{E}[X_i^2] = \frac{1}{2}$, we obtain:*

$$Var(S_+) = Var(S_-) = n \cdot \left( \frac{1}{2} - \left( \frac{1}{\sqrt{2\pi}} \right)^2 \right) = n \cdot \left( \frac{1}{2} - \frac{1}{2\pi} \right).$$

*In summary, the means and variances of $S_+$ and $S_-$ are:*

$$\mathbb{E}[S_+] = n \cdot \frac{1}{\sqrt{2\pi}}, \quad \mathbb{E}[S_-] = -n \cdot \frac{1}{\sqrt{2\pi}},$$

$$Var(S_+) = Var(S_-) = n \cdot \left( \frac{1}{2} - \frac{1}{2\pi} \right).$$

*Proof.* Using the above lemma, we now prove the theorem. The distributions of non-generalizable interactions are defined as the sum of positive and negative interactions, *i.e.*, $\mathbf{A}^{(m),+} = S_+$ and $\mathbf{A}^{(m),-} = S_-$. Therefore, the means and variances of $\mathbf{A}^{(m),+}$ and $\mathbf{A}^{(m),-}$ are:

$$\mathbb{E}[\mathbf{A}^{(m),+}] = \binom{n}{m} \cdot \frac{\sigma}{\sqrt{2\pi}}, \quad \mathbb{E}[\mathbf{A}^{(m),-}] = -\binom{n}{m} \cdot \frac{\sigma}{\sqrt{2\pi}},$$

$$Var(\mathbf{A}^{(m),+}) = Var(\mathbf{A}^{(m),-}) = \binom{n}{m} \cdot \left( \frac{1}{2} - \frac{1}{2\pi} \right) \sigma^2.$$

This demonstrates that the expectation distributions of non-generalizable interactions exhibit a spindle-shaped pattern, as required.

$\square$

## F  EXTRACTING THE SPARSEST AND-OR INTERACTIONS

Recent studies Li & Zhang (2023); Chen et al. (2024) have introduced a method to simultaneously extract **AND interactions** $I_S^{\text{AND}}$ and **OR interactions** $I_S^{\text{OR}}$ from the network output. Given a masked sample $\boldsymbol{x}_T$, the method decomposes the output $v(\boldsymbol{x}_T)$ into two components:

$$v(\boldsymbol{x}_T) = u_T^{\text{AND}} + u_T^{\text{OR}}, \tag{14}$$

where $u_T^{\text{AND}}$ represents the contribution of **AND interactions**, and $u_T^{\text{OR}}$ represents the contribution of **OR interactions**. Specifically, the decomposition is defined as:

$$u_T^{\text{AND}} = 0.5 \cdot v(\boldsymbol{x}_T) + \gamma_T, \quad u_T^{\text{OR}} = 0.5 \cdot v(\boldsymbol{x}_T) - \gamma_T, \tag{15}$$

where $\{\gamma_T : T \subseteq N\}$ is a set of learnable parameters that control the decomposition. Based on Theorem 2.1, the interactions are computed as:

$$I_S^{\text{AND}} = \sum_{T \subseteq S} (-1)^{|S|-|T|} u_T^{\text{AND}}, \quad I_S^{\text{OR}} = -\sum_{T \subseteq S} (-1)^{|S|-|T|} u_{N \setminus T}^{\text{OR}}. \tag{16}$$

To ensure sparsity in the extracted interactions, the parameters $\{\gamma_T\}$ are optimized by minimizing a **LASSO-like loss function**:

$$\min_{\{\gamma_T\}} \sum_{S \subseteq N} \left| I_S^{\text{AND}} \right| + \left| I_S^{\text{OR}} \right|. \tag{17}$$

**Handling Small Noises**  Small noises $\delta_S$ in the network output can significantly affect the extracted interactions, particularly for high-order interactions. To address this, Li & Zhang (2023) proposed incorporating a noise term $\delta_S$ into the decomposition:

$$u_T^{\text{AND}} = 0.5 \cdot (v(\boldsymbol{x}_T) - \delta_T) + \gamma_T, \quad u_T^{\text{OR}} = 0.5 \cdot (v(\boldsymbol{x}_T) - \delta_T) - \gamma_T. \tag{18}$$

Here, $\{\delta_T\}$ and $\{\gamma_T\}$ are jointly optimized by minimizing the loss function in Eq. equation 17. The values of $\{\delta_T\}$ are constrained to the interval $[-\zeta, \zeta]$, where $\zeta = 0.02 \cdot \mathbb{E}_{\boldsymbol{x}} |v(\boldsymbol{x}) - v(\boldsymbol{x}_\emptyset)|$.

**Algorithm for Interaction Extraction**  The complete procedure for extracting AND-OR interactions is summarized in Algorithm 1.

---

**Algorithm 1** Extraction of AND-OR Interactions

---

**Require:** Input sample $\boldsymbol{x} = [x_1, x_2, \ldots, x_n]$, DNN output function $v(\boldsymbol{x})$, threshold $\tau$, noise bound $\zeta$
**Ensure:** AND interactions $I_S^{\text{AND}}$, OR interactions $I_S^{\text{OR}}$, salient interaction sets $\Omega^{\text{AND}}, \Omega^{\text{OR}}$
 1: Initialize learnable parameters $\{\gamma_T\}$ and $\{\delta_T\}$ for all $T \subseteq N$
 2: $\zeta \leftarrow 0.02 \cdot \mathbb{E}_{\boldsymbol{x}} |v(\boldsymbol{x}) - v(\boldsymbol{x}_\emptyset)|$ {Compute noise bound}
 3: **while** not converged **do**
 4:     **for** each subset $T \subseteq N$ **do**
 5:         Compute masked sample $\boldsymbol{x}_T$
 6:         $u_T^{\text{AND}} \leftarrow 0.5 \cdot (v(\boldsymbol{x}_T) - \delta_T) + \gamma_T$ {Decompose into AND component}
 7:         $u_T^{\text{OR}} \leftarrow 0.5 \cdot (v(\boldsymbol{x}_T) - \delta_T) - \gamma_T$ {Decompose into OR component}
 8:     **end for**
 9:     **for** each subset $S \subseteq N$ **do**
10:         $I_S^{\text{AND}} \leftarrow \sum_{T \subseteq S} (-1)^{|S|-|T|} u_T^{\text{AND}}$ {Compute AND interaction}
11:         $I_S^{\text{OR}} \leftarrow -\sum_{T \subseteq S} (-1)^{|S|-|T|} u_{N \setminus T}^{\text{OR}}$ {Compute OR interaction}
12:     **end for**
13:     Optimize $\{\gamma_T\}$ and $\{\delta_T\}$ by minimizing $\sum_{S \subseteq N} |I_S^{\text{AND}}| + |I_S^{\text{OR}}|$ {LASSO-like loss}
14:     Constrain $\delta_T \in [-\zeta, \zeta]$ for all $T \subseteq N$
15: **end while**
16: $\Omega^{\text{AND}} \leftarrow \{S \subseteq N : |I_S^{\text{AND}}| < \tau\}$ {Select salient AND interactions}
17: $\Omega^{\text{OR}} \leftarrow \{S \subseteq N : |I_S^{\text{OR}}| < \tau\}$ {Select salient OR interactions}
18: **return** $I_S^{\text{AND}}, I_S^{\text{OR}}, \Omega^{\text{AND}}, \Omega^{\text{OR}}$

---

## G  EXPERIMENTS: NOISE INJECTION LEADS TO THE NON-GENERALIZABLE INTERACTIONS IN THE SPINDLE-SHAPED DISTRIBUTION

In this subsection, we revised a well-trained DNN by injecting non-generalizable representations into it to to obtain a DNN with non-generalizable features (The detailed experimental settings are the same in the Section 3.4 of the main paper). To examine the newly emerged interactions, we conducted two experiments, revising the well-trained DNN from two perspectives.

In the first experiment, we revised the well-trained DNN by adding Gaussian noises $\epsilon \sim \mathcal{N}(0, \sigma^2)$ to the network parameters of a well-trained DNN. Then, we computed the interactions in the well-trained DNN, denoted by $I_S^{\text{AND}}$ and $I_S^{\text{OR}}$, and the interactions in the revised DNN, denoted by $I_{S,\text{noise}}^{\text{AND}}$ and

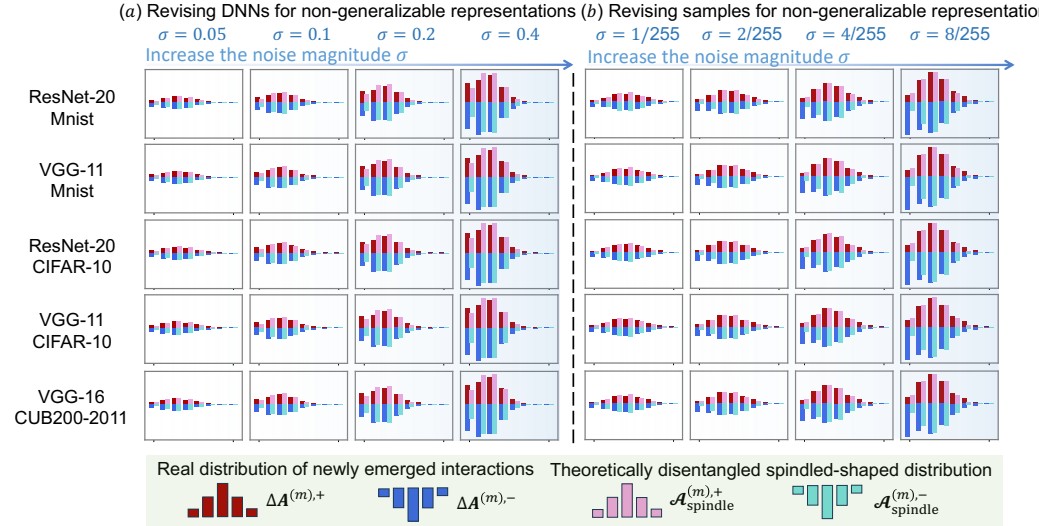

Figure 7: The distributions of newly emerged interactions ($\Delta\mathbf{A}^{(m),+}, \Delta\mathbf{A}^{(m),-}$). All newly emerged interactions followed a spindle-shaped distribution. The magnitude of newly emerged interactions increased along with the increase of the injected non-generalizable representations. The theoretical estimated distributions of interactions ($\mathcal{A}_{\text{spindle}}^{(m),+}$ and $\mathcal{A}_{\text{spindle}}^{(m),-}$) in Equation (7) well matched the true distributions of newly-emerged interactions.

$I_{S,\text{noise}}^{\text{OR}}$. Thus, $\Delta I_S^{\text{AND}} = I_S^{\text{AND}} - I_{S,\text{noise}}^{\text{AND}}$ and $\Delta I_S^{\text{OR}} = I_S^{\text{OR}} - I_{S,\text{noise}}^{\text{OR}}$ represented the interaction effects newly emerged in the revised DNN. We measured the distribution of newly emerged interactions in the revised DNN $\Delta\mathbf{A}^{(m),+}$ and $\Delta\mathbf{A}^{(m),-}$ by following Equation (6).

$$\Delta\mathbf{A}^{(m),+} = \sum_{S:|S|=m} \max\{\Delta I_S^{\text{AND}}, 0\} + \max\{\Delta I_S^{\text{OR}}, 0\}$$
$$\Delta\mathbf{A}^{(m),-} = -\sum_{S:|S|=m} \min\{\Delta I_S^{\text{AND}}, 0\} + \min\{\Delta I_S^{\text{OR}}, 0\} \tag{19}$$

In the second experiment, given a specific sample $x$, we revised this sample by adding adversarial perturbations[7] and generate $x_{\text{noise}}$, as the injection of non-generalizable representations. Then, we used the same setting in the first experiment to measure the distribution of interactions newly emerged in the revised sample.

We conducted experiments on various DNN architectures, including ResNet-20, VGG-11, and VGG-16, using datasets such as MNIST, CIFAR-10, and CUB200-2011. Figure 7 shows the average distribution of interactions newly emerged across different samples in the two experiments. All newly emerged interactions exhibited a spindle-shaped distribution. Notably, the spindle-shaped distribution became more pronounced as the injected noise strength increased. These results partially validate the hypothesis that non-generalizable interactions follow a spindle-shaped distribution.

Furthermore, we used the proposed analytical formulation for the spindle-shaped distribution ($\mathcal{A}_{\text{spindle}}^{(m),+}$ and $\mathcal{A}_{\text{spindle}}^{(m),-}$, following Equation (7)) to fit the newly emerged non-generalizable interactions. The results showed an excellent fit, providing additional evidence for the reliability of our theoretical framework.

# H  EXPERIMENTS: DISENTANGLING THE DISTRIBUTIONS OF GENERALIZABLE AND NON-GENERALIZABLE INTERACTIONS IN CORRECTLY CLASSIFIED SAMPLES

In Section 2.3 , we have proposed and anaylzed two claims: generalizable interactions exhibit a decaying pattern, while non-generalizable interactions often exhibit a spindle-shaped pattern. In

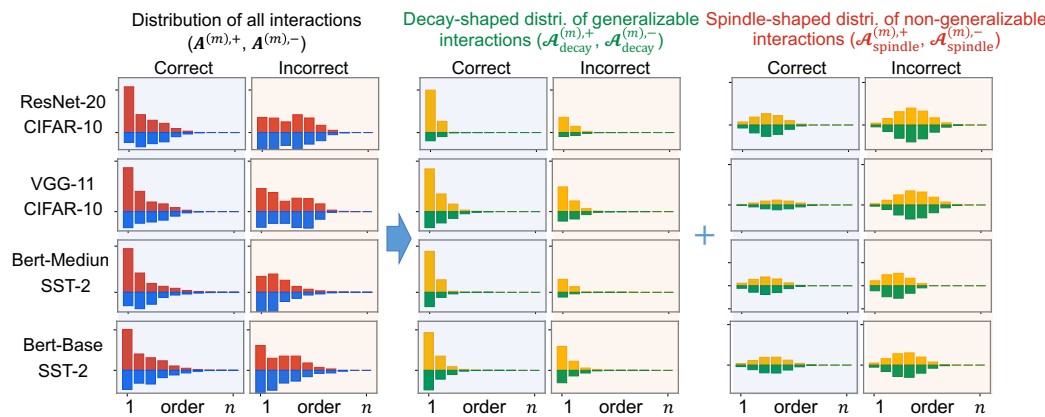

Figure 8: Distribution of generalizable interactions ($\mathcal{A}_{\text{decay}}^{(m),+}$ and $\mathcal{A}_{\text{decay}}^{(m),-}$) and that of non-generalizable interactions ($\mathcal{A}_{\text{spindle}}^{(m),+}$ and $\mathcal{A}_{\text{spindle}}^{(m),-}$) in correctly and misclassified samples. In misclassified samples, the strength of decaying interactions (generalizable interactions) is relatively low. In correctly classified samples, the strength of decaying interactions is significantly higher, while spindle-shaped interactions (non-generalizable interactions) persist.

Section 3, we further investigate and derive the mathematical expressions for these two distributions. Additional experiments demonstrate that our theory accurately captures the true distributions of generalizable and non-generalizable interactions. This provides a novel mathematical perspective, enabling us to directly assess the quality of neural network representations on given samples, *i.e.*, identify representation defects. We find that even in correctly classified samples, neural networks still model non-generalizable interactions that exhibit a spindle-shaped pattern.

Specifically, we trained VGG-11 and ResNet-20 on the CIFAR-10 dataset. We also trained BERT-Medium and BERT-Base models on the SST-2 dataset. Given test samples from the neural networks, we computed their distributions ($\mathbf{A}^{(m),+}$ and $\mathbf{A}^{(m),-}$). Furthermore, we employed a disentanglement method to extract the distributions of generalizable interactions ($\mathcal{A}_{\text{decay}}^{(m),+}$ and $\mathcal{A}_{\text{decay}}^{(m),-}$) and non-generalizable interactions ($\mathcal{A}_{\text{spindle}}^{(m),+}$ and $\mathcal{A}_{\text{spindle}}^{(m),-}$), following Equation (6).

Figure 8 indicates that in misclassified samples, the strength of the decaying interactions (i.e., generalizable interactions) is relatively low, suggesting that the model does not effectively capture the generalizable representations in these samples, leading to poor classification performance. In contrast, in correctly classified samples, the strength of the decaying interactions is significantly higher than in misclassified samples, indicating that the model indeed captures the generalizable representations. However, spindle-shaped interactions (i.e., non-generalizable interactions) still exist, suggesting that the model continues to model non-generalizable interactions even in correctly classified samples.

## I  EXPERIMENTS VERIFICATION OF THEOREM 3.1

**Formulation of the matrix** $M(\delta)$**.** According to (Ren et al., 2024b), the matrix $M(\delta)$ is calculated as $M(\delta) = (\boldsymbol{J}^\top \boldsymbol{J} + 2^n \text{diag}(\boldsymbol{c}))^{-1} \boldsymbol{J}^\top \boldsymbol{J}$, where $\boldsymbol{J} \stackrel{\text{def}}{=} [\boldsymbol{J}(\boldsymbol{x}_{S_1}), \boldsymbol{J}(\boldsymbol{x}_{S_2}), \cdots, \boldsymbol{J}(\boldsymbol{x}_{S_{2^n}})]^\top \in \mathbb{R}^{2^n \times 2^n}$ is a matrix to represent the triggering values of $2^n$ interactions (*w.r.t.* $2^n$ columns) on $2^n$ masked samples (*w.r.t.* $2^n$ rows). $\boldsymbol{x}_{S_1}, \boldsymbol{x}_{S_2}, \cdots, \boldsymbol{x}_{S_{2^n}}$ enumerate all masked samples. The vector $\boldsymbol{c} \stackrel{\text{def}}{=} \text{vec}(\{\text{Var}[\epsilon_T] : T \subseteq N\}) = \text{vec}(\{2^{|T|}\sigma^2 : T \subseteq N\}) \in \mathbb{R}^{2^n}$ denotes the variances of the triggering values of $2^n$ interactions.

Theorem 3.1 shows that the ubiquitous uncertainty in network parameters and training data will eliminate the mutually counteracting high-order interactions, thereby avoiding. We conducted the following experiments to verify the Theorem 3.1. Specifically, We trained VGG-11, VGG-16, AlexNet on the CIFAR-10 dataset and CUB200-2011 dataset, respectively. We also trained BERT-Medium models on the SST-2 datasets. We calculated the distributions $\mathbf{A}^{(m),+}, \mathbf{A}^{(m),-}$ of interactions extracted form the three timepoints in the DNN's overfitting phase. Given the extremely overfitting

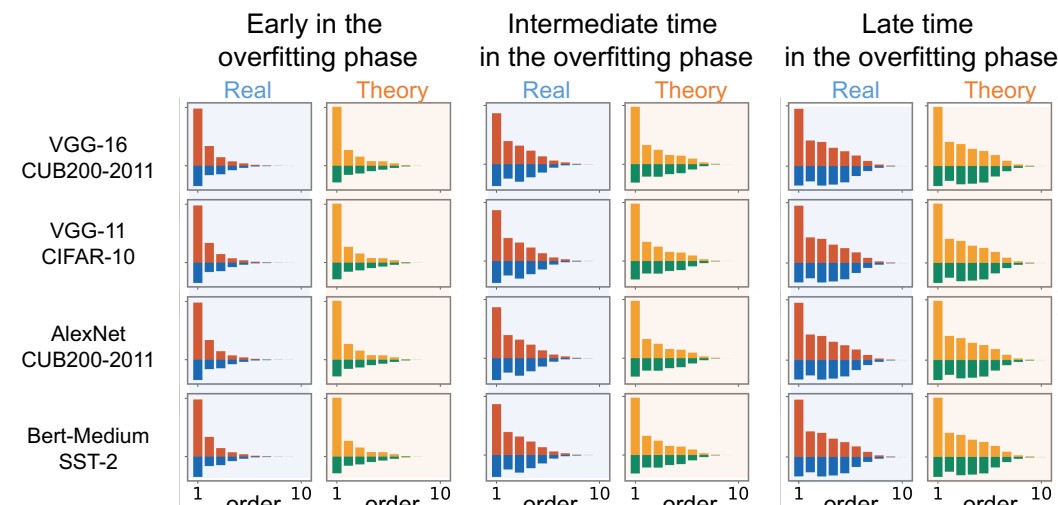

Figure 9: Comparison between the theoretical distribution of interactions ($\mathcal{A}_{\text{decay}}^{(m),+}$ and $\mathcal{A}_{\text{decay}}^{(m),-}$) and the real distribution of interaction strength ($\mathbf{A}^{(m),+}$ and $\mathbf{A}^{(m),-}$) in the overfitting phase.

DNN, we used the Theorem 3.1 to predict the dynamics os interactions of different orders before the overfitting phase. We verified whether the Theorem 3.1 could well predict the real dynamics of interaction strength of different orders in real DNNs.

Figure 9 shows that the theoretical distribution ($\mathcal{A}_{\text{decay}}^{(m),+}$ and $\mathcal{A}_{\text{decay}}^{(m),-}$) of interactions well matched the real distribution ($\mathbf{A}^{(m),+}$ and $\mathbf{A}^{(m),-}$) of interactions in the overfitting phase. This verifies that the Theorem 3.1 could well predict the dynamics of interactions of different orders in real DNNs. In this way, we could remove the eliminate the mutually counteracting high-order interactions by setting a enough large uncertainty level $\delta$ (In our experiments, we set $\delta \leq 1e - 3$).

## J EXPERIMENTS: PENALIZING NON-GENERALIZABLE INTERACTIONS IMPROVES THE DNN'S PERFORMANCE

To bridge the gap between theoretical analysis and practical application, we performed a follow-up experiment aimed at enhancing the performance of DNNs by penalizing non-generalizable interactions encoded by the DNN during training. The core idea involves using a dedicated validation set to identify and suppress non-generalizable interactions while minimizing the classification loss in an end-to-end training framework.

Our method comprises two main stages. First, to assess the generalizability of learned interactions, we trained a baseline DNN on a separate validation set (the experimental details are introduced in Section 2.2). This allowed us to quantify which interactions were non-generalizable. Second, we designed a loss function that incorporates a penalty term based on this quantification. During the main model's training, this loss function simultaneously minimizes the classification error and penalizes the absolute magnitudes of the identified non-generalizable interactions, denoted as $I_S^{\text{non}}$.

We evaluated the proposed approach on the CIFAR-10 dataset using a VGG-11 architecture. The results confirm the effectiveness of our method: the proportion of generalizable interactions, rose significantly from $28\%$ to $39\%$, while non-generalizable interactions decreased from $72\%$ to $61\%$.

In terms of model accuracy, the enhanced VGG-11 model achieved a test accuracy of $85.2\%$ on CIFAR-10, an improvement of $1.7\%$ over the baseline ($83.5\%$). The approach exhibited even stronger gains in data-scarce settings. For example, on an object recognition task with only 200 training samples using ResNet-18, our method improved classification accuracy by $6.0\%$ (from $76.6\%$ to $82.6\%$).

In conclusion, the experiment validates that penalizing non-generalizable interactions not only elevates the quality of the learned representations but also boosts model performance, particularly under limited training data conditions.

# K EXPERIMENTAL DETAILS

## K.1 MODELS AND DATASETS

We trained a variety of deep neural networks (DNNs) on multiple datasets. For image classification tasks, ResNet-20 and VGG-11 were trained on the MNIST dataset (licensed under Creative Commons Attribution-Share Alike 3.0), while ResNet-20, VGG-11, and VGG-16 were trained on the CIFAR-10 dataset (MIT license). Additionally, AlexNet and VGG-16 were trained on the CUB-200-2011 dataset (license unspecified). For natural language processing tasks, BERT-Medium and BERT-Base were trained on the SST-2 dataset (license unspecified).

For the CUB-200-2011 dataset, we followed standard practices (Ren et al., 2024b; Zhang et al., 2024) to crop images and remove background regions using the bounding boxes provided by the dataset. These cropped images were resized to $224 \times 224$ and input into the DNNs. For the Tiny ImageNet dataset, due to computational constraints, we selected 50 classes from the total 200 classes at equal intervals (*i.e.*, the 4th, 8th, ..., 196th, and 200th classes). All images were resized to $224 \times 224$. For the MNIST dataset, images were resized to $32 \times 32$ for classification.

In experiments investigating the two-stage dynamics of interactions during the DNN learning process, we followed the same experiment settings in (Ren et al., 2024b) to introduce a small proportion of label noise to the CIFAR-10 and CUB-200-2011 datasets to highlight the relationship between higher-order interactions and overfitting. Specifically, 1% of the training samples in the MNIST and CIFAR-10 datasets were randomly selected and their labels reset. Similarly, 5% of the training samples in the CUB-200-2011 dataset were randomly selected and their labels reset.

## K.2 DETAILS ON COMPUTING INTERACTIONS

First, we provide a summary of the mathematical settings of the hyper-parameters for interactions in Table 1, including the scalar output function of the DNN $v(\cdot)$, the baseline value $\boldsymbol{b}$ for masking, and the threshold $\tau$. These settings are uniformly applied to all DNNs. More detailed settings for different datasets can be found below.

**Image data.** For image data, we considered image patches as input variables to the DNN. To generate a masked sample $\boldsymbol{x}_S$, we followed Zhang et al. (2024) to mask the patch on the intermediate-layer feature map corresponding to each image patch in the set $N \setminus S$. Specifically, we considered the feature map after the second ReLU layer for VGG-11/VGG-16 and the feature map after the first ReLU layer for AlexNet. For the VGG models and the AlexNet model, we uniformly partitioned the feature map into $8 \times 8$ patches, randomly selected 10 patches from the central $6 \times 6$ region (*i.e.*, we did not select patches that were on the edges), and considered each of the 10 patches as an input variable in the set $N$ to calculate interactions. We considered each of the 10 patches as an input variable in the set $N$ to calculate interactions. We used a zero baseline value to mask the input variables in the set $N \setminus S$ to obtain the masked sample $\boldsymbol{x}_S$.

**Natural language data.** We considered the input tokens as input variables for each input sentence. Specifically, we randomly selected 10 words that are meaningful (*i.e.*, not including stopwords, special characters, and punctuations) as input variables in the set $N$ to calculate interactions. We used the "mask" token with the token id 103 to mask the tokens in the set $N \setminus S$ to obtain the masked sample $\boldsymbol{x}_S$.

For all DNNs and datasets, we randomly selected 50 samples from the testing set to compute interactions, and averaged the interaction strength of the $k$-th order on each sample to obtain $I_{\text{real}}^{(k)}$.

## K.3 NORMALIZATION METHODS

In the experiment of analyzing the distribution of interactions at the different timepoints in the training process of DNNs, we need to normalize the interaction strength of the $k$-th order on each

| Output function $v(\cdot)$ | $v(\boldsymbol{x}) = \log \frac{p(y^{\text{truth}}|\boldsymbol{x})}{1-p(y^{\text{truth}}|\boldsymbol{x})}$ |
|---|---|
| Threshold $\tau$ | $\tau = 0.02\,\mathbb{E}_{\boldsymbol{x}}[|v(\boldsymbol{x}) - v(\boldsymbol{x}_{\emptyset})|]$ |
| Baseline value $\boldsymbol{b}$ | Image data: using the zero baseline on the feature map after ReLU |
| | Text data: using the [MASK] token |

Table 1: Mathematical setting of hyper-parameters for interactions.

sample. Specifically, for interactions of each $k$-th order, we normalized the strength of salient interactions as $\tilde{I}_S^{\text{AND}} = I_S^{\text{AND}}/Z$ and $\tilde{I}_S^{\text{OR}} = I_S^{\text{OR}}/Z$, where $Z$ is the averaged sum of the absolute values of all salient 1-order interactions across the all samples, *i.e.*, $Z = E_x \sum_{S \in \Omega^{\text{AND}}, |S|=1} |I_S^{\text{AND}}| + E_x \sum_{S \in \Omega^{\text{OR}}, |S|=1} |I_S^{\text{OR}}|$. The normalization removes the effect of the explosion of output values during the training process and enables us to only analyze the relative distribution of interaction strength.

### K.4 COMPUTE RESOURCES

All DNNs can be trained within 12 hours on a single NVIDIA GeForce RTX 3090 GPU (with 24G GPU memory). Computing all interactions on a single input sample usually takes 35-40 seconds, which is acceptable in real applications.

## L  MORE EXPERIMENTAL RESULTS

### L.1  MORE RESULTS FOR THE DISTRIBUTIONS OF GENERALIZABLE AND NON-GENERALIZABLE INTERACTIONS IN SECTION 2.3

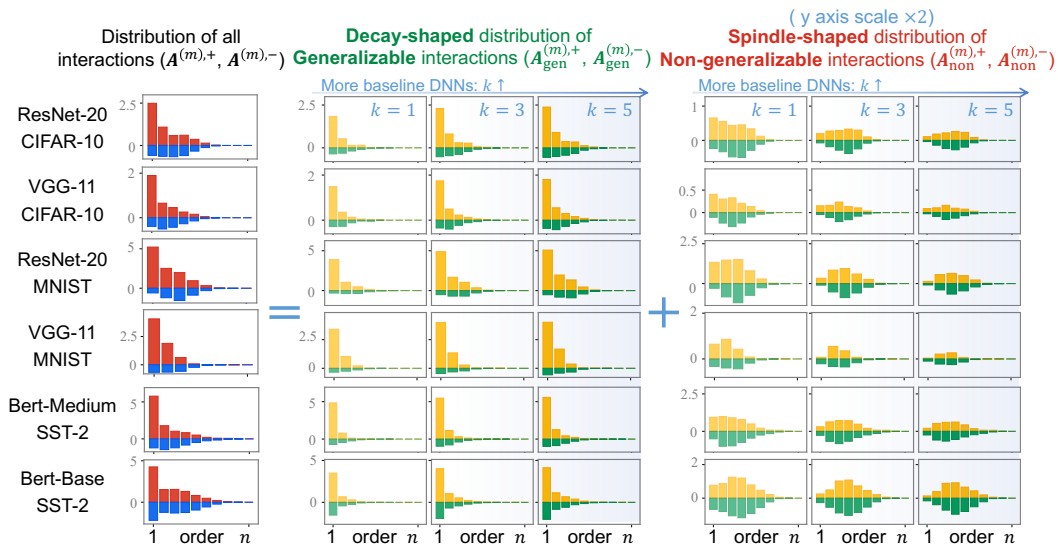

Figure 10: Generalizable interactions ($\mathbf{A}_{\text{gen}}^{(m),+}$ and $\mathbf{A}_{\text{gen}}^{(m),-}$) exhibit a decay-shaped distribution, while non-generalizable interactions ($\mathbf{A}_{\text{non}}^{(m),+}$ and $\mathbf{A}_{\text{non}}^{(m),-}$) exhibit a spindle-shaped distribution. Furthermore, ablation studies show that when using three baseline DNNs (*i.e.*, $k = 3$), the two distributions have already converged (close to those observed when $k = 5$). To enhance clarity, we double the y-axis scale for non-generalizable interactions.

### L.2  MORE RESULTS FOR THE TWO-STAGE DYNAMICS OF INTERACTIONS IN SECTION 2.3

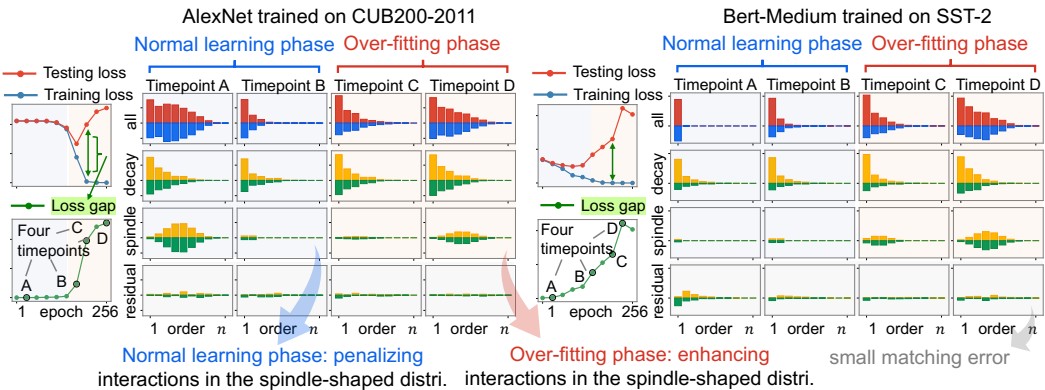

Figure 11: The two-stage dynamics of interactions in the learning process of a DNN. The first row illustrates the change of interactions ($\mathbf{A}^{(m),+}$ and $\mathbf{A}^{(m),-}$) throughout the entire learning process. The remaining three rows visualize the distributions of the disentangled generalizable interactions ($\mathcal{A}_{\text{spindle}}^{(m),+}$ and $\mathcal{A}_{\text{spindle}}^{(m),-}$), non-generalizable interactions ($\mathcal{A}_{\text{decay}}^{(m),+}$ and $\mathcal{A}_{\text{decay}}^{(m),-}$), and the residual term ($\mathcal{A}_{\text{residual}}^{(m),+}$ and $\mathcal{A}_{\text{residual}}^{(m),-}$).

L.3   MORE RESULTS FOR THE DISENTANGLED DISTRIBUTIONS IN SECTION 3.4

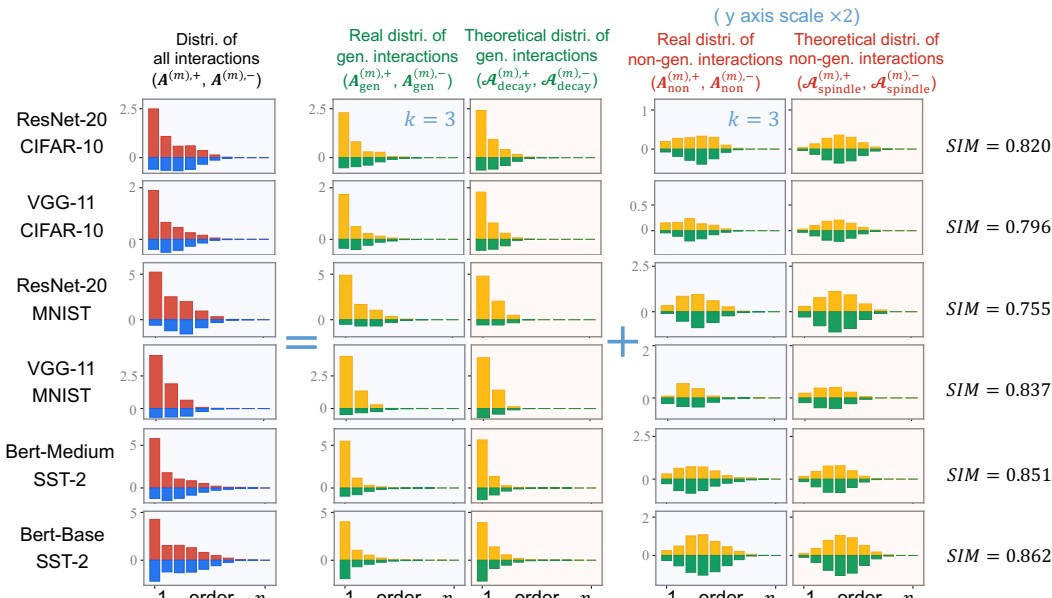

Figure 12: Verification of our proposed method 's accuracy in capturing the true distributions of generalizable interactions and non-generalizable interactions. The figure demonstrates a high degree of alignment between the theoretical disentangled distributions (represented by $\mathcal{A}_{\text{decay}}^{(m),+}$, $\mathcal{A}_{\text{decay}}^{(m),-}$, $\mathcal{A}_{\text{spindle}}^{(m),+}$, $\mathcal{A}_{\text{spindle}}^{(m),-}$) and the real distributions (represented by $\mathbf{A}_{\text{gen}}^{(m),+}$, $\mathbf{A}_{\text{gen}}^{(m),-}$, $\mathbf{A}_{\text{non}}^{(m),+}$, $\mathbf{A}_{\text{non}}^{(m),-}$), as evidenced by the consistently high Jaccard similarity values ($SIM$). To enhance clarity, we double the y-axis scale for non-generalizable interactions.

## L.4 MORE RESULTS FOR THE NOISE INJECTIONS EXPERIMENTS IN SECTION 3.4

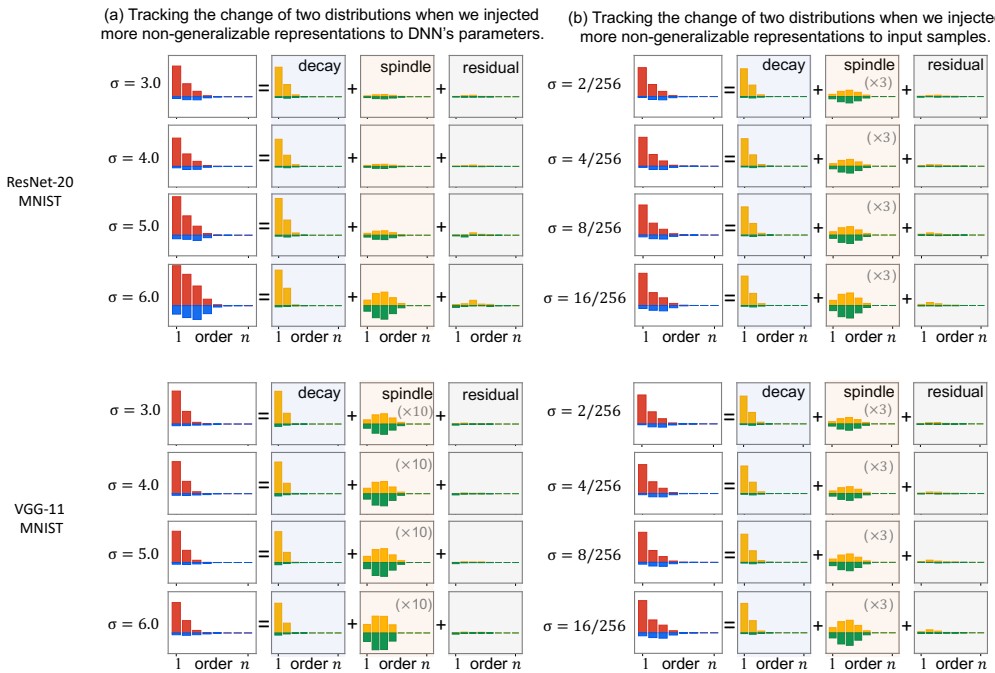

Figure 13: As we increase the noise magnitudes $\sigma^2$, *i.e.*, injecting more non-generalizable representations into the DNN, the significance of interactions in the spindle-shaped distribution ($\mathcal{A}_{\text{spindle}}^{(m),+}$ and $\mathcal{A}_{\text{spindle}}^{(m),-}$) increases. In contrast, interactions in the decay-shaped distribution ($\mathcal{A}_{\text{decay}}^{(m),+}$ and $\mathcal{A}_{\text{decay}}^{(m),-}$) remain largely unaffected. This observation further validates the faithfulness of our method.

## M THE USE OF LARGE LANGUAGE MODELS (LLMS)

This paper employs large language models (LLMs) solely for partial polishing of wording and phrasing, with the aim of enhancing the clarity and standardization of the writing.

