# OpenReview forum: "Formulating Generalizable and Non-Generalizable Interactions in DNNs"
_ICLR.cc/2026/Conference — Submitted to ICLR 2026_

### Official Review · Reviewer_PyG7 · 2025-10-25

**Soundness:** 3
**Presentation:** 1
**Contribution:** 2
**Rating:** 4
**Confidence:** 3

**Summary:**

This paper analyses the generalisation behaviour of deep neural networks (DNNs) by representing inputs as symbolic variables and examining their compositional interactions. The authors theoretically and empirically demonstrate that generalizable interactions exhibit a low-order, decay-shaped distribution, whereas non-generalizable interactions follow a spindle-shaped distribution, concentrated in mid-order terms. They propose a disentangling framework to separate these distributions and validate its fidelity across multiple DNN architectures.

**Strengths:**

1. Novel Perspective: The paper provides a novel perspective on generalisation and provides theoretical support for notions such as simplicity bias. The ideas are well-motivated and appear to productively build upon prior works.
2. Empirical Validation: The paper provides empirical validation (using ResNet, VGG, and BERT architectures) of its theoretical contributions and utilises them to illustrate the different phases of training.
3. Contribution to Prior Works and Notions: The authors make clear connections to prior works and notions such as overfitting.

**Weaknesses:**

1. Utilisation of baseline DNNs: Using deep neural networks trained on the test set to provide the ground-truth is not ideal. These baseline DNNs have the potential to over-fit and thus provide misleading results. It is suggested to use multiple baselines to compensate for the fact that baselines may not be able to capture all generalising patterns. However, this would not solve the issue of overfitting, resulting in noisy results, especially as only one baseline needs to provide a positive result for a pattern to be deemed generalising. This is evident by the fact that increasing the number of baselines to five leads to worse performance compared to using three baselines.
2. Overstating the Practical Applicability: It is mentioned that the proposed methods could be used to reduce overfitting. Considering all interactions for an input is combinatorial in the number of elements in the input. In the experiments, relatively small-dimensional input data is considered, meaning enumerating these interactions is feasible; however, scaling this to large input dimensions does not seem feasible.

**Questions:**

1. Did you consider using majority voting for the baselines, rather than just requiring the generalizability of one baseline?
2. What are the computational complexities of your methods, such as the one to disentangle the generalisable and non-generalisable distributions?
3. Did you explore the effectiveness of regularisation techniques on the observed distributions? Perhaps it is the case that weight-decay already has the intended effect on the distributions that utilising your method would have.

---

### Official Review · Reviewer_Z8tb · 2025-10-28

**Soundness:** 2
**Presentation:** 3
**Contribution:** 2
**Rating:** 6
**Confidence:** 2

**Summary:**

This paper investigates the generalization power of Deep Neural Networks (DNNs) through the lens of "interactions"—specifically, AND/OR logical relationships between input variables that the model uses for inference. The core claim is that a DNN's generalization can be explained by the generalizability of these underlying interactions. The authors make two key empirical observations: Generalizable interactions follow a decay-shaped distribution, and non-generalizable interactions (linked to overfitting) follow a spindle-shaped distribution. Building on these observations, the paper proposes a method to mathematically formulate and disentangle these two distributions within a given DNN. Experimental results on various models and datasets are presented to show that the theoretically disentangled distributions align well with empirically measured ones.

**Strengths:**

1.	Novel and Intriguing Perspective: The paper tackles the fundamental problem of generalization from a fresh and mechanistic angle. Moving from a holistic view of the model to analyzing the generalizability of its constituent "interaction primitives" is a conceptually interesting and potentially powerful approach.
2.	Theoretical Formulation: The authors go beyond mere empirical observation by providing mathematical formulations for the two proposed distributions (decay and spindle). The attempt to create a disentanglement method (Equation 10) is a non-trivial contribution.
3.	Comprehensive Evaluation: The paper is thorough in its experimental validation, testing the claims across multiple architectures (CNNs, Transformers) and data modalities (vision, language). The experiments on tracking interactions during training (two-stage dynamics) and under noise injection provide supporting evidence for their claims.

**Weaknesses:**

1. Insular Research Framework: A major concern is the high degree of self-citation and the lack of independent validation. The entire theoretical foundation of "symbolic generalization" and interaction extraction is built almost exclusively upon a series of papers from the same research group. This creates a significant risk of an "echo chamber" effect, where a complex framework is developed without being stress-tested, adopted, or critically evaluated by the broader community. The validity of the entire pipeline hinges on the acceptance of their prior work, which has yet to become mainstream.

2. Questionable Practical Utility and Impact: The paper demonstrates that it can describe a phenomenon (the distributions), but the "so what?" remains unclear. The proposed applications (e.g., identifying representation quality without test data, early stopping) are presented as potential future work or in a preliminary form. It is uncertain whether this complex analysis provides a tangible advantage over simpler, established metrics for detecting overfitting or estimating generalization. The framework feels heavy, and it's difficult to see a wide range of researchers or practitioners adopting it.

3. Technical Complexity and Assumptions: The method relies on several strong assumptions and heuristic settings (e.g., the formulation of M(\delta) in Theorem 3.1, the scaling parameter \alpha in the spindle distribution). While the experiments show a good fit, the underlying derivations may not be fully rigorous or universally applicable. The computational cost of extracting interactions for all subsets of input variables, though noted as "acceptable," is non-trivial and scales poorly, limiting its use on large-scale problems.

4. Circularity in Validation: The method for quantifying "ground-truth" generalizable interactions (using transferability to baseline DNNs trained on test samples) feels somewhat circular. It risks validating the method against a benchmark that is constructed using the same underlying principles and may share similar biases.

**Questions:**

See the weaknesses.

---

### Official Review · Reviewer_gEfX · 2025-10-30

**Soundness:** 1
**Presentation:** 2
**Contribution:** 1
**Rating:** 2
**Confidence:** 4

**Summary:**

Based on an "explanation logical model" of a DNN/LLM, the paper tries to analyze the generalizable and non-generalizable modules of the logical model.

**Strengths:**

Sorry, I don't find any strength

**Weaknesses:**

1: The paper is built upon two key statements: (1) the AND–OR logical model **explains**, and (2) is **equivalent to**, DNNs and/or LLMs. However, neither statement is correct.

Examining the definition of the AND–OR logical model $h(x)$ in Eq.3, the expressions of and/or interactions $I_S^{AND}$ and $I_S^{OR}$ as well as $u_T^{AND}$ and $u_T^{OR}$ (in Theorem 2.1), reveals that the logical model $h(x_{mask})$ is actually expressed as logical combinations of $v(x_T)$’s, which are the **outputs** of the DNN/LLM for $x_T$’s. In other words, the DNN/LLM $v$ takes a raw input $x_T$ and outputs a $v(x_T)$, whereas the logical model takes the **outputs** $v(x_T)$ as its inputs, rather than the raw inputs $x_T$. Thus, the logical model and the DNN/LLM do not operate on the same level: the logical model cannot replace the DNN/LLM, and it is inaccurate to claim that the two are equivalent (as suggested in Figure 1) or that one explains the other.

According to the paper’s formulation, the logical model merely takes a set of $v(x_T)$s and produces one such  $v(x_T)$, effectively analyzing the logical relationships among different DNN/LLM outputs. However, the paper misinterprets this process as indicating model equivalence and explanatory power.

1(a): Figure 1 (as well as other figures in the paper) is misleading in the sense that each of the token in the logical model is not a raw token $x_T$, but is the output $v(x_T)$ which has been processed by the DNN/LLM. The authors should correct this to avoid confusion.

2: I am skeptical about the nontriviality of Theorem 2.1. From Eq. (3) and the definitions of the related quantities, $h(x_T)$ can be written as logical combinations of $v(x_{T’})$’s, where $T’\subset N$. If we expand the logical combination expressions of $h(x_T)$ in terms of $v(x_{T’}$s,  I am wondering if all the expressions reduces to the simple and trivial expression of $h(x_T) = v(x_T)$. If that is the case, that means the AND-OR logical model implements merely a trivial logic – the identity function, and the theorem would be trivial. Based on my reading of the proof, this seems likely. I recommend that the authors explicitly expand these expressions—for instance, those illustrated in Figure 1—to demonstrate whether the logical model and Theorem 2.1 are indeed nontrivial.


3: Most theory of the paper, basically the only two theorems (Theorem 2.1, 3.1), are from prior works  and not contributions of this submission.

I am happy to increase the score if the above concerns are addressed.

**Questions:**

No further questions

---

### Official Review · Reviewer_nYLM · 2025-11-06

**Soundness:** 1
**Presentation:** 2
**Contribution:** 2
**Rating:** 2
**Confidence:** 3

**Summary:**

This paper investigates the generalization behavior of deep neural nets from the perspective of "symbolic interactions”, a concept from prior work where a network's inference logic is decomposed into a set of AND/OR operations on input variables (e.g. tokens or image patches).
The authors' main claim is that a DNN's generalization power can be explained by the properties of these interactions. They propose two main claims:
1. Generalizable interactions (those that are useful for classification on the test set) follow a "decay-shaped" distribution over their "order", meaning most generalizable logic is of low-order.
2. Non-generalizable interactions (spurious correlations) follow a "spindle-shaped" distribution, with most being of medium-order order.
The paper's main contribution is a method to disentangle these two distributions from a trained DNN.

**Strengths:**

- The paper is overall well-written and easy to follow.
- The paper offers a novel perspective on generalization.
- The goal of identifying non-generalizable components without extensive test data is a plus.

**Weaknesses:**

- My main problem is with how the paper checks if its own method is correct.
To prove that the theory works, the authors first need to know which logic patterns (or "interactions") are actually good (generalizable) and which are bad (non-generalizable). This set of "good" patterns is their "ground truth."
But, they don't use a standard or direct way to find this ground truth. Instead, they use a workaround (a "proxy"):
1. They train their main model on the training data.
2. Then, they train several other "baseline" models using only the test data.
3. They decide a logic pattern is "generalizable" only if it shows up in both their main model and at least one of the "baseline" models.
This is a weird definition for "ground truth." The paper doesn't prove that this method—checking if a pattern transfers to a model trained on test data—is actually a reliable way to measure generalization.

- As the authors admit, this explanatory power of this framework is only on the level of interaction-order, which is rather coarse-grained.

- The theoretical model for the 'decay' shape is a significant concern. It isn't derived from first principles of generalization but is instead an empirical model of a process (noise-injection), known to encourage a spindle shape (by suppressing higher order components). This makes the overall disentanglement model feel circular. For me, this raises the concern that the clean separation is an artifact of these specific definitions, rather than a fundamental property of the network's learned logic. In my view, the findings in this paper cannot refute the claims of Zhou et al. 2024 (which asserts that non-generalizing components are of both medium and high order).

**Questions:**

- See weaknesses.
- Can you reproduce the decay + spindle phenomenon even if no noise-injection step is performed?

Typos:
- 393 typo “thedemonstrates”

---

### Meta-Review · Area_Chair_LJNH · 2025-12-27

**Summary:**

This paper studies the generalization of deep neural networks by analyzing the generalization properties of symbolic interactions. Most reviewers lean toward rejection because they find the empirical and/or theoretical support for the paper's claims insufficient. Some theoretical results are also viewed as somewhat trivial. More importantly, several reviewers are not convinced that the AND–OR logical model provides a faithful explanation of DNN generalization, noting that a shift in meaning across the argument may occur throughout the paper. Unfortunately, the authors did not submit a rebuttal to address these key concerns.

**Reviewer Concerns:**

No rebuttal was submitted, so the reviewers' concerns remain unaddressed.

**Reviewer Scores:**

Since the authors did not submit a rebuttal, it is very unlikely that any of the reviewers will change their score.

---

### Decision · Program_Chairs · 2026-01-26

Reject